# A CCL2+DPP4+ subset of mesenchymal stem cells expedites aberrant formation of creeping fat in humans

Fengfei Wu [1], Fangting Wu [1], Qian Zhou [1], Xi Liu [1], Jieying Fei [1], Da Zhang [1], Weidong Wang [1], Yi Tao [1], Yubing Lin [1], Qiaoqiao Lin [1], Xinghua Pan [2], Kai Sun [3], Fang Xie [1,4] ✉ & Lan Bai [1,4] ✉

Creeping fat is a typical feature of Crohn's disease. It refers to the expansion of mesenteric adipose tissue around inflamed and fibrotic intestines and is associated with stricture formation and intestinal obstruction. In this study, we characterize creeping fat as pro-adipogenic and pro-fibrotic. Lipidomics analysis of Crohn's disease patients (sixteen males, six females) and healthy controls (five males, ten females) reveals abnormal lipid metabolism in creeping fat. Through scRNA-seq analysis on mesenteric adipose tissue from patients (five males, one female) and healthy controls (two females), we identify a CCL2+DPP4+ subset of mesenchymal stem cells that expands in creeping fat and expedites adipogenic differentiation into dystrophic adipocytes in response to CCL20+CD14+ monocytes and IL-6, leading to the formation of creeping fat. Ex vivo experiments (tissues from five males, one female) confirm that both CCL20+CD14+ monocytes and IL-6 activate DPP4+ mesenchymal stem cells towards a pro-adipogenic phenotype. This study provides a comprehensive investigation of creeping fat formation and offers a conceptual framework for discovering therapeutic targets for treatment of Crohn's disease.

A fibrotic and stricturing phenotype develops in more than half of patients with Crohn's disease (CD), and up to 80% of these patients require surgical intervention[1]. Moreover, CD frequently recurs after resection[2]. Despite the increasing diversity of anti-inflammatory therapies, there are still limitations in reducing the incidence of strictures in patients with CD[1]. Potent antifibrotic therapies are still an unmet medical need.

Creeping fat (CF), a unique extra-intestinal phenomenon in patients with CD, expands and wraps around sites of intestinal fibrosis, stricture and inflammation[3]. CF and intestinal strictures often co-occur, and both tend to be extremely fibrotic. The formation of CF involves a complex interplay among multiple cell lineages, including mesenchymal, endothelial and immune cells spatially located alongside adipocytes, termed the adipogenic niche[4,5]. Notably, adipocytes have a smaller cell size, higher flattening and dystrophic signature with extracellular matrix (ECM) deposition and poly-partitioned lipid droplets surrounding them, where the immune cells and stromal vascular fraction (SVF) cells accumulate.

Although CF was first published in 1932[6], the cause of the increased number and aberrant lipid metabolic signature in dystrophic adipocytes during CF formation remains unclear. Recent elegant investigations suggested that translocation of the gut microbiota into the mesentery and fibronectin production by activated intestinal muscle cells could be responsible for CF[3,7]. Given that CF increases in

[1]Guangdong Provincial Key Laboratory of Gastroenterology, Institute of Gastroenterology of Guangdong Province, Department of Gastroenterology, Nanfang Hospital, Southern Medical University, Guangzhou, China. [2]Department of Biochemistry and Molecular Biology, School of Basic Medical Sciences, Southern Medical University, and Guangdong Provincial Key Laboratory of Single Cell Technology and Application, Guangzhou, China. [3]Department of General Surgery, Nanfang Hospital, Southern Medical University, Guangzhou, China. [4]These authors jointly supervised this work: Fang Xie, Lan Bai.
✉ e-mail: stellaff@126.com; bailan_99@yeah.net

size principally depending on hyperplasia [the formation of new adipocytes through the differentiation of resident mesenchymal stem cells (MSCs)][8], we hypothesized that there is a prominent subset of MSCs distinctively expanding in CF and expediting de novo adipogenesis in response to inflammatory and fibrotic intestinal segments affected by CD which is in direct spatial and functional contact with the immediately overlying CF.

Here, we resolved the cellular composition and phenotype of the adipogenic niche in human mesenteric adipose tissue (MAT) collected from surgical resections and identified a CCL2+DPP4+ subset of MSCs (MSC1-S1s), with both pro-adipogenic and pro-fibrotic signatures and an extreme increase that distinguishes CF. Lipidomics profiling of MAT across CF, adjacent MAT and healthy tissue controls, combined with the results of single-cell metabolism (scMetabolism), characterized CF as hypometabolic, surprisingly with a rich series of activated pro-adipogenic differentiation and dysfunctional lipid metabolic pathways. Ex vivo validation in CF-derived primary cells suggested that MSC1-S1s accumulate in CF and expedite adipogenic differentiation into committed preadipocytes and then dystrophic adipocytes responding to CCL20+ classical monocytes and IL-6 assessed by flow cytometry, driving aberrant formation of CF in humans.

## Results

### Histopathology and lipidomics of the hyperplastic MAT adipogenic milieu in CD

To explore the nature of CF, we investigated it by histopathology. CF was patchy rather than continuous, with extended finger-like projections gripping the diseased segments of intestine (Fig. 1a), which we observed to be a consistent feature in the CD cohort that consists of 60 patients who underwent surgical resections due to strictures from CD (Supplementary Data 1). The presence of CF was observed in 59 out of 60 patients. We collected 133 strictured intestinal segments from these patients and found the presence of CF in 118 of them. We observed increased intestinal segment thickness and MAT expansion in fibrotic regions of patients with CD (Supplementary Fig. 1a). To investigate the relationship between the amount of CF and the severity of fibrosis, we performed correlation analysis and observed a statistically significant correlation ($r = 0.8337$, $p = 0.0023$, Supplementary Fig. 1b) between the maximal thickness of MAT and the maximal thickness of the small intestine in patients with CD. Of note, CF encroached into the intestinal muscularis and contained smaller adipocytes and abundant infiltrated immune cells (Fig. 1b and Supplementary Fig. 1c). Using scanning electron microscopy (SEM), we observed a significant increase in smaller and nascent adipocytes, fibre disarray and ECM deposition in CF compared with CD-MAT and H-MAT (Fig. 1c, d).

We performed lipidomics by using whole-adipose tissue to first assess whether an aberrant lipid metabolic signature could be detected in MAT and, if so, whether this was unique to CF, where adipocytes shrink with immune cell infiltration and nascent adipocytes emerge with ECM deposition and surrounded by poly-partitioned lipid droplets. For the CF ($n = 20$, fifteen males and five females), paired CD-MAT ($n = 18$, thirteen males and five females) and H-MAT ($n = 15$, five males and ten females), the whole metabolic profiles of CF and H-MAT are presented as a heatmap (Fig. 1e and Supplementary Fig. 1d), which showed distinct metabolite profiles for CF. Out of the 963 metabolites detected, 198 were identified as significantly differential metabolites, with 34 upregulated and 164 downregulated in CF compared with H-MAT (Supplementary Data 2 and 3). KEGG pathway enrichment analysis of differential metabolites showed aberrant glycerolipid metabolism and regulation of lipolysis in adipocytes in CF compared with H-MAT (Supplementary Fig. 1e, f). The rates of lipolysis may be related to the diameter of adipocytes[9]. Furthermore, key genes enriched in the glycerolipid metabolism could stimulate the proliferation, differentiation, and lipolysis of MSCs[10].

### Dynamic reconstruction of the cellular milieu in CF

Given the degree of aberrant lipid metabolic activity across our patients, we comprehensively resolved the cellular milieu of the adipogenic niche in human MAT by profiling stromal vascular fraction (SVF) cells from CF ($n = 6$, five males and one female), paired CD-MAT ($n = 3$, all males) and H-MAT ($n = 2$, both females) using 10X Genomics Chromium droplet scRNA-seq (Fig. 1f and Supplementary Data 4). The human MAT SVF single-cell atlas included 68,174 unsorted cells (filtered to the 32,953 highest-quality cells), which were clustered based on differential expression of marker genes and visualized using a uniform manifold approximation and projection (UMAP) plot (Fig. 2a, b). Clustering analysis revealed thirteen clusters: MSCs, endothelial cells (ECs), pericytes (PCs), macrophages (Macs), monocytes, neutrophils, B cells, dendritic cells (DCs), plasma cells, mast cells, natural killer (NK) cells, CD8+ T cells (T cell 1) and CD4+ T cells (T cell 2) (Fig. 2c and Supplementary Data 5). When the thirteen clusters were distinguished by tissue source, the clusters belonging to ECs and PCs consisted almost entirely of cells from CF (Fig. 2b). We observed other dramatic changes in cells in CD; for example, MSCs constituted on average 49% of all cells in H-MAT but only ~14% in CF and/or CD-MAT. We also observed an increase in monocytes and neutrophils and a reduction in macrophages in CF (Supplementary Fig. 2a). This comparison to control tissues revealed that CF is clearly defined by an abundance of distinct stromal cells and major innate immune cells. To confirm these changes, we utilized MAT from an independent cohort of ten control subjects (seven males, three females) and eight individuals with CD (six males, two females). Flow cytometry experiments corroborated the results of scRNA-seq analysis, supporting the identification and frequency of stromal cells (Supplementary Fig. 2b, c and Supplementary Data 6).

We further investigated cluster-specific gene signatures (Supplementary Data 7) and found a localized signature enriched for the regulation of the innate immune response in myeloid cells; similarly, MSCs were highly enriched for ECM organization, ECs for angiogenesis, and PCs for muscle contraction (Fig. 2d). The results of scRNA-seq transcriptional activity of CF and H-MAT revealed that of the 35,038 genes detected, 3886 were identified as significantly differentially expressed genes (DEGs), with 1946 upregulated and 1940 downregulated in CF (Supplementary Data 8). GO analysis of upregulated genes highlighted strong signatures in CF for fibrosis, angiogenesis and innate immune response (Fig. 2e and Supplementary Data 9). The majority (1967) of identified DEGs were differentially expressed within a single cluster, with only 27 genes significantly differentially expressed in more than eight clusters, such as MSC, myeloid cell and T-cell populations. We posit that this result reflects differential pathway activity in MSCs in patients with CD, where we observed high levels of costimulation of collagen formation, ECM degradation, metabolism of fat-soluble vitamins and basement membranes (Supplementary Fig. 2d). Using adipogenesis gene sets, we computed a gene set score for every cell in each cluster, which showed high levels of adipogenesis in MSCs, stromal cells and myeloid cells (Supplementary Fig. 2e and Supplementary Data 10), suggesting that monocytes/macrophages and ECs may have cooperative effects on the adipogenic differentiation of MSCs in CF formation. The scRNA-seq data for CD-MAT vs. H-MAT and CF vs. CD-MAT are shown in Supplementary Data 11 and 12.

We next sought to explore the transcriptomic landscape of CD genetic susceptibility loci within the MAT at single-cell level. We found an overall higher enrichment of genome-wide association study (GWAS) genes[11] from CD-associated loci in MSCs, ECs, macrophages, monocytes and neutrophils (Supplementary Fig. 2f). High and specific expression of genes, including *FBLN1* (rs2238823), *COL5A1* (rs11103429), *CAVIN1* (rs11871801), *CCL2* (multiple risk alleles) and *CPEB4* (multiple risk alleles) (Supplementary Data 13), are associated with fibrosis[12–14], which implies that the cell clusters that specifically express these genes are involved in fibrosis.

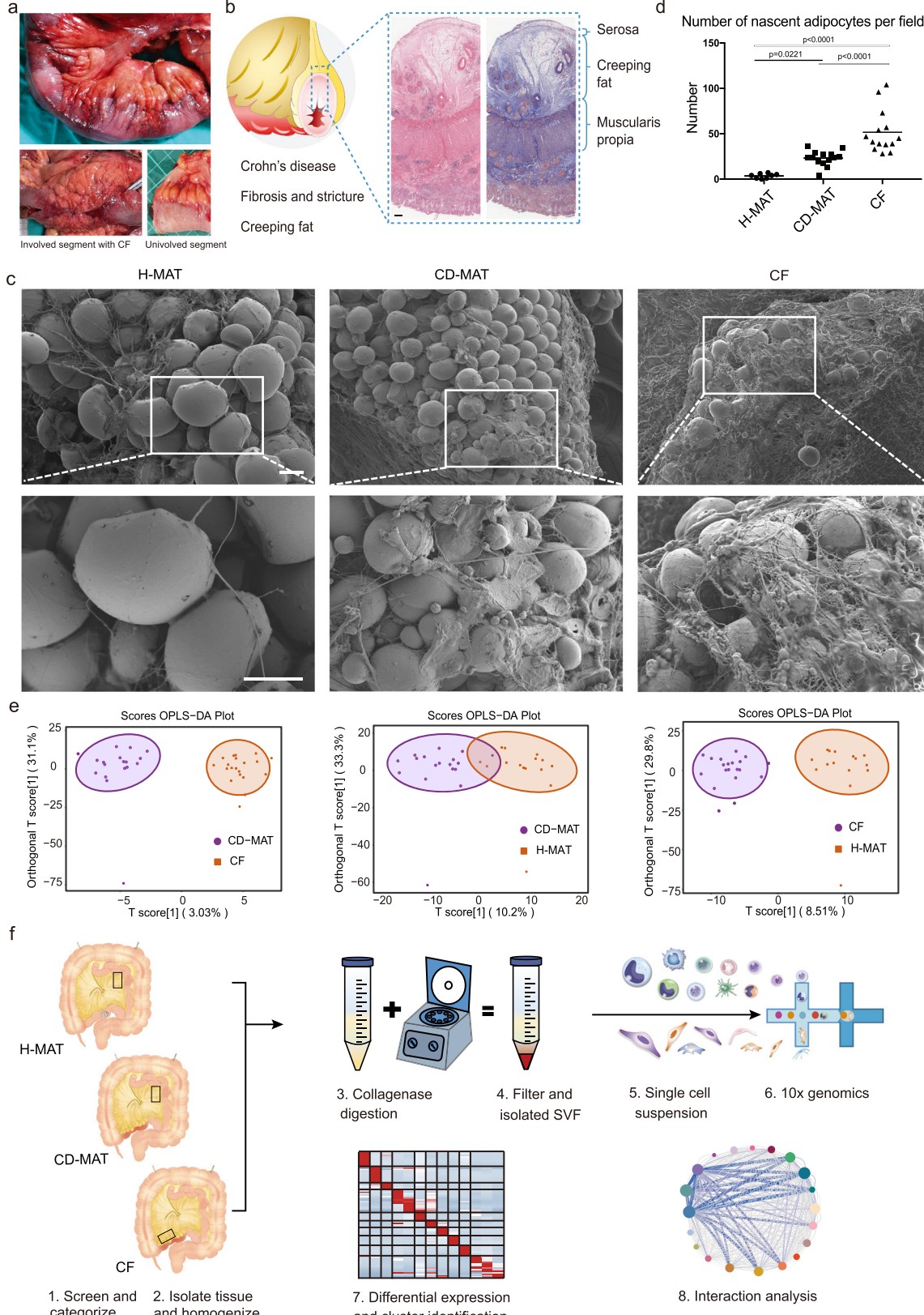

## MSC subpopulations are prone to adipogenesis in CF

Previous studies have suggested that MSCs in subcutaneous adipose tissue orchestrate both the expansion and elimination of adipose tissue[15,16]. Based on our high-resolution MSC map in human MAT, we identified three MSC subpopulations (MSC1-3) (Fig. 3a, b). The canonical mesenchymal markers *PDGFRA* and *THY1* were expressed predominantly in MSCs (Supplementary Fig. 3a). MSC1s were marked by *DPP4* and *BMP8B* but did not express adipocyte markers (Fig. 3c and Supplementary Fig. 3b). MSC2s expressed *SFRP4* and *F3* (also known as *CD142*). MSC3s expressed *ICAM1* (also known as *CD54*), *APOE*, and several adipocyte identified genes (Fig. 3c, Supplementary Fig. 3b, c and Supplementary Data 14). In addition, tissue staining and flow

**Fig. 1 | Characteristics, lipidomics and scRNA survey of human MAT in CD.**
**a** Macroscopic characteristics of representative involved ileal segments with
attached hyperplastic CF from CD patients. **b** Haematoxylin and eosin- and Masson
trichrome-stained ileal segments with attached CF. Scale bars, 500 μM. **c** Scanning
electron microscopy examination of H-MAT, CD-MAT and CF. Scale bars, 50 μM.
**d** Absolute number of nascent adipocytes (diameter < 10 μM) in H-MAT of control
subjects ($n = 8$), CD-MAT and CF of CD patients ($n = 14$) counted in (**c**). The centre
line represents the mean and $p$ value were determined via two-sided one-way
ANOVA. **e** Orthogonal partial least squares discriminant analysis (OPLS-DA) score

plot of whole mesenteric adipose tissue in CD-MAT vs. CF, CD-MAT vs. H-MAT and
CF vs. H-MAT ($n = 15$–20 per group). Differentially expressed lipid metabolites were
detected based on ultra-performance liquid chromatography tandem mass spec-
trometry (UPLC–MS/MS). **f** Schematic of the experimental design for scRNA
sequencing. Human mesenteric adipose tissue samples were isolated from H-MAT,
CD-MAT and CF, dissociated into single-cell suspensions, and analysed using 10×
Genomics Chromium droplet scRNA sequencing. Cells were clustered via differ-
ential gene expression and ligand–receptor analysis was performed to assess
interactions among cell types. Source data are provided as a Source data file.

---

cytometry experiments confirmed these cells as MSCs (Fig. 3d and
Supplementary Fig. 3d).

Considering the potential role of MSCs in CF formation, we eval-
uated the discrepancies in the three subpopulations in CF, CD-MAT
and H-MAT and found that MSC3s were more abundant in CF and CD-
MAT than in H-MAT (Supplementary Fig. 3e), whereas MSC1s were less
abundant and negatively correlated with the mesenteric creeping fat
index (MCFI; Supplementary Fig. 3f)[17]. To functionally validate the
existence and frequency of identified MSC subpopulations, we
recruited an additional cohort of ten individuals (four control subjects,
three males and one female and six patients with paired CF and CD-
MAT samples, five males and one female) (Supplementary Data 6). We
used a panel of markers for flow cytometry analysis, which included
CD45[+] immune cells, CD31[+] endothelial cells, CD31[−]CD45[−]CD146[+]
pericytes and CD31[−]CD45[−]CD146[−] MSCs. From these MSCs, we purified
the CD26[+]CD142[−]CD54[−] MSC1s, CD142[+]CD26[−]CD54[−] MSC2s and
CD54[+]CD26[−]CD142[−] MSC3s. The results corroborated the scRNA-seq
analysis, supporting the identification of MSC subpopulations and
expansion of MSC3s in CF and CD-MAT (Fig. 3e).

We next assessed the individual and combined expression of
commonly used lineage marker genes (Supplementary Data 15). To
fully reveal the substantial discrepancy in adipogenic potential among
the three subpopulations, we evaluated the ex vivo adipogenic ability
of MSC1-3s. All three subpopulations underwent robust adipocyte
differentiation and activated adipocyte-specific genes when treated
with the standard complete cocktail of adipogenic inducers (Fig. 3f, g).
However, under minimal adipogenic conditions, where the cocktail
contained only a low concentration of insulin, MSC2s and MSC3s still
differentiated efficiently into adipocytes, whereas MSC1s displayed
very low adipogenic capacity. In addition, MSC1s displayed enhanced
competence for proliferation and differentiation into osteocytes and
chondrocytes compared with MSC2s and MSC3s (Fig. 3h and Supple-
mentary Fig. 3g). Together, MSC1s, which function as highly pro-
liferative multipotent progenitors, were decreased in CF and CD-MAT
compared with H-MAT. In contrast, MSC2 and MSC3 cells, which are
relatively restricted to the adipocyte lineage, were dramatically
increased in CF and CD-MAT. MSC3s might be committed pre-
adipocytes that express adipocyte identified genes and are poised to
generate many lipid droplets with minimal stimulation. Accordingly,
these data suggest that MSC subpopulations in CF compared with
H-MAT were in a vibrant adipogenic state and had higher adipogenic
ability.

### MSC subpopulations expedite adipogenesis in CF

To explore the hierarchy in CF undergoing de novo adipogenesis, we
visualized the transcriptional profile of MSC subpopulations, mapped
them along a pseudotemporal trajectory and interrogated their
directionality via Monocle pseudotime analysis. These analyses pre-
dicted a clear developmental trajectory of MSC1s to MSC3s, and
MSC2s appeared to represent an intermediate state (Fig. 4a). We
unexpectedly discovered two subsets of MSC1s with different tran-
scriptional profiles, named MSC1 subset 1 (MSC1-S1) and MSC1 subset 2
(MSC1-S2) (Fig. 4a). To investigate the essence of MSC1s, we visualized
the transcriptional profiles of MSCs in different groups and found that

MSC1-S1s were predominant in CF and CD-MAT while MSC1-S2s were
prevalent in H-MAT but absent in CF and CD-MAT (Fig. 4b). Gene
profiling of MSC1-S1s, combined with flow cytometry analysis, revealed
that CCL2 (also known as MCP1) expression could distinguish MSC1-
S1s and validated the accumulation of MSC1-S1s in CF compared with
H-MAT (Fig. 4c, d). MSC1-S1s highly expressed *PIM1* and *IL-6*, which
might promote adipocyte differentiation (Fig. 4c)[18]. Together, we
extrapolated that these cells have a distinct effect on adipogenesis.

Analysis of DEGs along MSC differentiation pseudotemporal tra-
jectories revealed that MSC1-S1s in CF showed accelerated differ-
entiation into MSC2s and MSC3s compared with MSC1-S2s in H-MAT
(Supplementary Fig. 4a and Supplementary Data 16). Due to intracel-
lular expression of MCP-1, we could not acquire live MSC1-S1s. Given
that MSC1-S1s mainly reside in CF and MSC1-S2s in H-MAT, as men-
tioned above, we next evaluated ex vivo adipogenic ability of MSC1s
derived from CF and H-MAT to mimic MSC1-S1s and MSC1-S2s. There
was no significant difference in adipogenic ability between these cells
when treated with complete adipogenic inducers (Fig. 4e). However,
under minimal adipogenic conditions, MSC1s in CF displayed higher
adipogenic ability than those in H-MAT and were surrounded by poly-
partitioned lipid droplets, similar to the unique manifestation in CF
samples from CD patients (Fig. 4f). More importantly, these differ-
ences were distinct in the early adipogenesis at day 6.

To elucidate the molecular mechanism and gene variety of MSC1-
S1 and MSC1-S2 adipogenesis, we assessed the GO enrichment of
MSC1-S1 DEGs and highlighted a suite of adipogenesis- and
inflammation-related gene networks (Supplementary Fig. 4b–d),
whereas MSC1-S2 DEGs were highly associated with fatty acid β-
oxidation. To further refine these analyses, we defined five gene co-
expression modules along the transition of MSC1-S1s or MSC1-S2s to
MSC2s and MSC3s. Module 2 was the dramatically upregulated module
during the differentiation of MSC1-S2s to MSC3s in H-MAT, which
represented metabolism-related genes (Fig. 4g). Modules 3–5 con-
tained multiple genes that were upregulated during the differentiation
of MSC1-S1s to MSC3s in CF. Module 3 contained multiple pro-
fibrogenic and pro-adipogenic genes and displayed ontology terms
that are consistent with the promotion of fibrosis and fat cell differ-
entiation (Fig. 4g, Supplementary Fig. 4e and Supplementary Data 17).
Moreover, modules 4 and 5 encompassed a group of upregulated
genes such as *HSPA12A*, a member of the heat shock protein family
required for adipocyte differentiation through positive feedback reg-
ulation with PPARγ[19], and IL-6, which is directly involved in promoting
adipogenesis and tissue fibrosis remodelling. Subsequently, flow
cytometry examination of the changes in cell surface biomarkers on
ex vivo isolated MSC1s (five males, one female) following IL-6 stimu-
lation demonstrated a preference for the transition from MSC1 to
MSC2 and MSC3 induced by IL-6 (Fig. 4h and Supplementary Fig. 5).

To investigate whether there are metabolic changes in the process
of MSC1-S1s accelerated adipogenesis, we evaluated all metabolism-
associated genes of MSCs by scMetabolism analysis and observed
lower metabolic activities in CF and CD-MAT compared with H-MAT
(Fig. 4i), indicating the presence of aberrantly deficient metabolism in
MSCs of CF and CD-MAT, the same results as those observed by lipi-
domics. Strikingly, analysis of lipid metabolites and associated genes

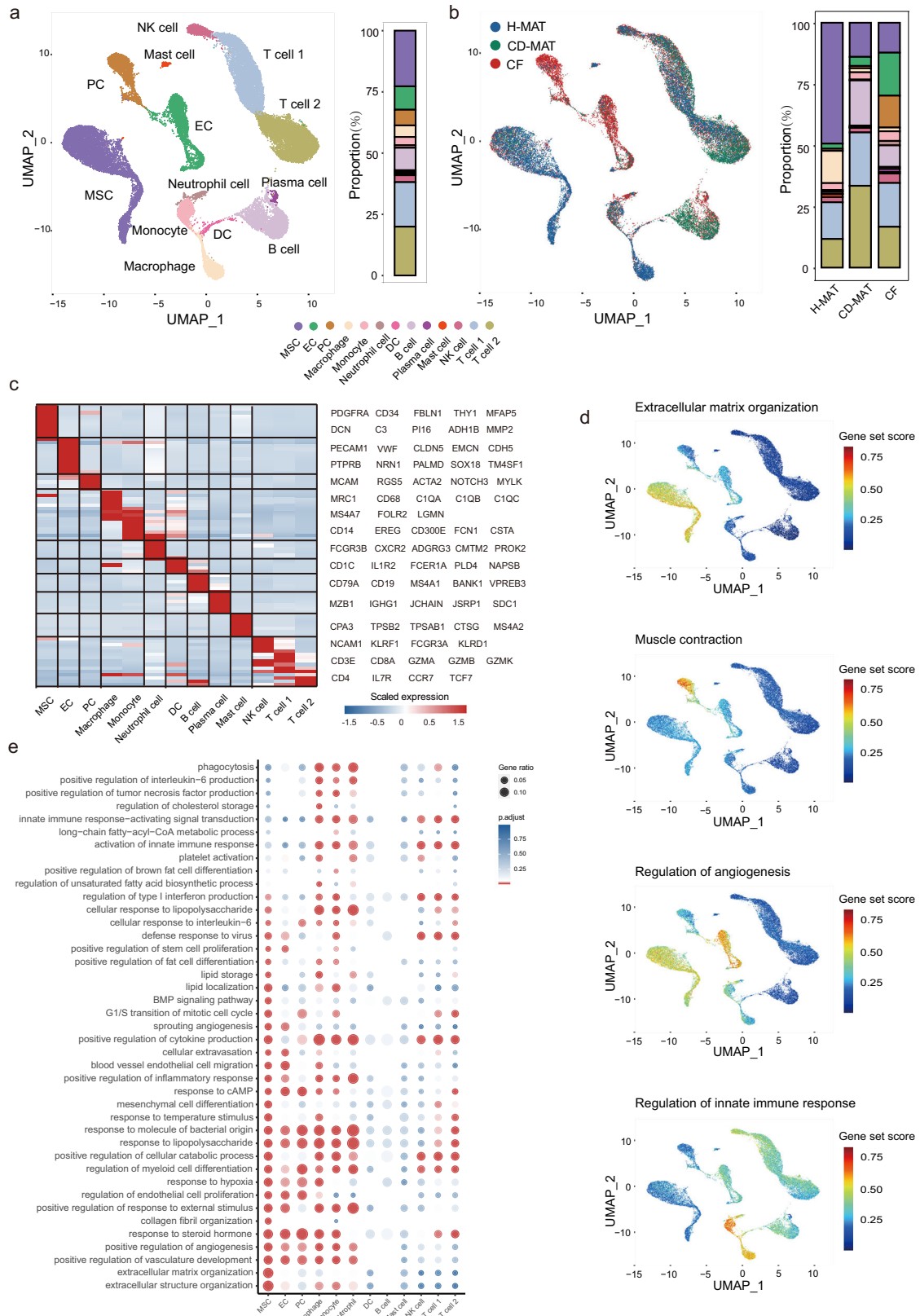

**Fig. 2 | Compositions and functions of the stromal vascular fractions in human MAT. a**, **b** scRNA-seq-generated UMAP plots of H-MAT, CD-MAT and CF distinguishing thirteen individual cell clusters (**a**) and tissue sources (**b**). Coloured bars indicate the frequency of cell types. **c** Heatmap showing scaled expression of specific marker genes for thirteen cell clusters. **d** Individual cell AUC score overlay for selected canonical pathway activities. Colour saturation indicates the gene set score. **e** scRNA-seq-generated dotplot heatmap showing GO terms enriched in thirteen cell cluster DEGs and *p* value were determined via two-sided Wilcoxon rank sum test. *p* < 0.05 was considered to indicate statistical significance.

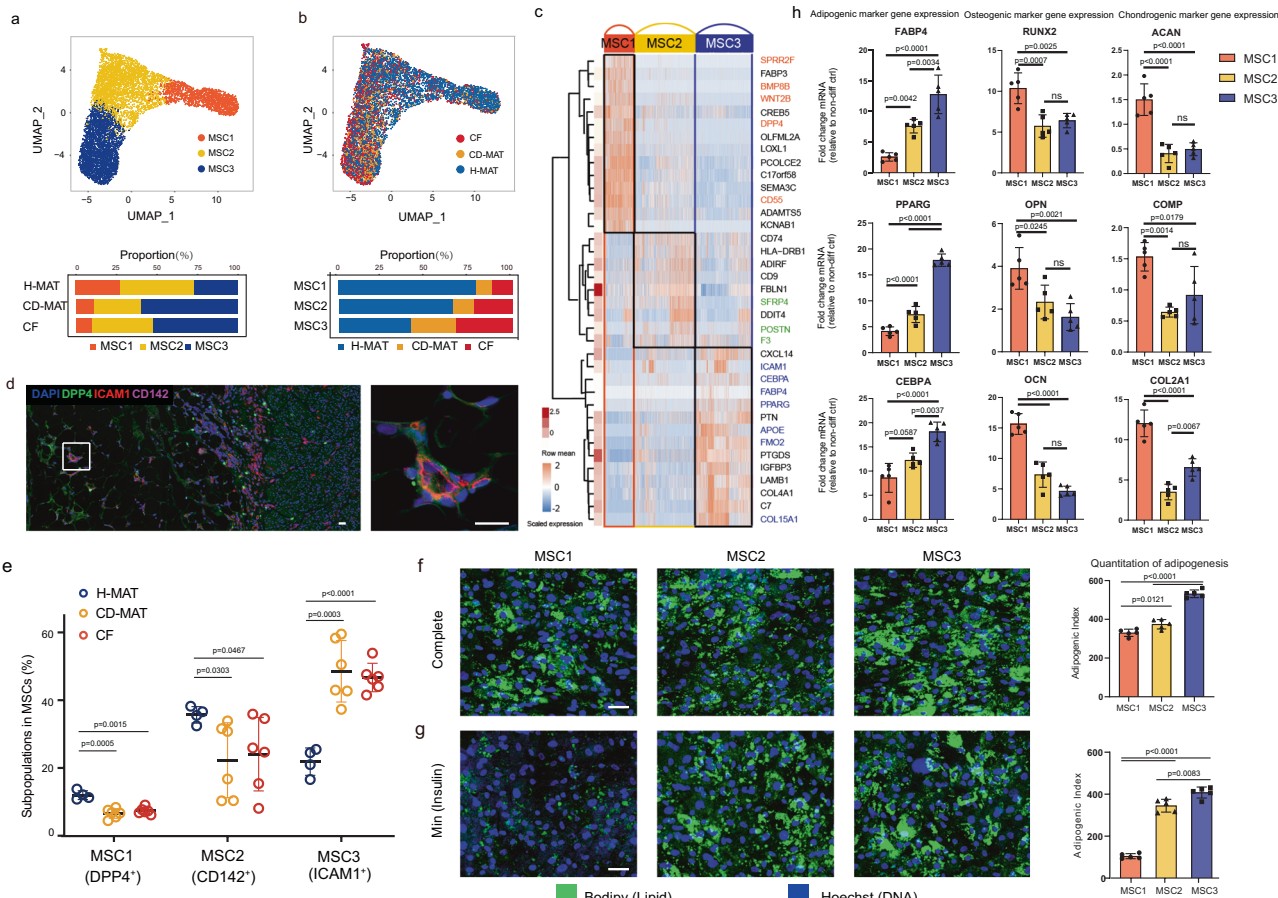

**Fig. 3 | Characterization of the genomics and function of MSC subpopulations.** **a**, **b** scRNA-seq-generated UMAP plots of MSCs distinguishing three individual subpopulations (**a**) and tissue sources (**b**). Coloured bars indicate the proportion of MSCs in H-MAT, CD-MAT and CF. **c** Heatmap (blue-to-red) showing the expression of specific marker genes for three MSCs; hierarchically clustered rows, mean gene expression (white-to-red, left). **d** Representative immunofluorescence images of DPP4 (green), ICAM1 (red) CD142 (pink) and DAPI (blue) in human MAT. Scale bars, 25 μM. **e** Flow cytometry showing the percentage of subpopulations in MSCs isolated from H-MAT, CD-MAT and CF of control subjects (*n* = 4) and CD patients

(*n* = 6). The centre line represents the mean. **f**, **g** Microscopy images of human MSC subpopulations after exposure to the complete adipogenic differentiation cocktail (**f**) or insulin only (Min) (**g**). [*n* = 5 biological replicates (BRs) per condition]. Scale bars, 50 μM. **h** mRNA levels of osteocyte-specific genes and chondrocyte-specific genes in MSC subpopulations exposed to osteogenic or chondrogenic differentiation inducers (representative of 5 BRs per condition). **e**, **f**–**h** The data are presented as means ± s.d. and *p* value were determined via two-sided one-way ANOVA. Source data are provided as a Source data file.

indicated upregulation of lipid metabolic pathways related to adipogenesis in MSC1-S1s, such as sphingolipid, arginine, nicotinate and nicotinamide metabolism pathways, which all aid in positively regulating adipogenic differentiation of MSCs (Supplementary Fig. 6)[20,21]. Altogether, these results suggested that accelerated adipogenesis during CF formation may be due to the increased differentiation of MSC1-S1s to MSC3s.

### Pro-inflammatory monocyte subset accumulates in CF and promotes MSC1-S1 adipogenesis

Given the abundance of data supporting a key role for monocytes and macrophages in adipose expansion[22], we sub-grouped myeloid cells into eight populations (Fig. 5a, b, Supplementary Fig. 7a, b and Supplementary Data 18). Neutrophils were identified by *FCGR3B* and *CXCR2* (Fig. 5b and Supplementary Fig. 7a, b), and neutrophils highly expressed *GCA*, which promoted adipogenesis of BMSCs[23], in CF (Supplementary Fig. 7c). Three macrophage and two DC subsets were defined (Fig. 5b and Supplementary Fig. 7d). Our analysis confirmed two types of monocytes: non-classical monocytes (ncMos) expressing *FCGR3A* and *HES4* and classical monocytes (cMos) expressing *NRG1* and *THBS1* (Fig. 5b and Supplementary Fig. 7a, b), where *THBS1* promotes monocyte adhesion to ECM in angiogenesis and modulates the

activation of TGF-β in colitis[24,25]. The cMos became increasingly of interest due to their high frequency and hypermetabolism in CF compared with H-MAT (Supplementary Fig. 7e, f).

To study the effect of cMos on adipogenesis, further clustering identified two subsets of cMos. cMo-1s, which highly expressed *CCL20* and *SLC7A5*, were mainly present in CF but absent in H-MAT (Fig. 5c–e and Supplementary Data 19). Assessment of the differential expression of transcription factors (TFs) (Supplementary Fig. 7g) and multiple genes (Fig. 5f) in cMo-1s led to the prediction that these cells promote fat cell differentiation, angiogenesis and response to lipopolysaccharide (Supplementary Fig. 7h). Focusing on the gene expression patterns along the transition between cMo subsets, pseudotime was reconstituted by trajectory analysis (Fig. 5g). Along the trajectory during the transition of cMo-2s to CF-enriched cMo-1s, *CXCL8*, *INHBA* and *CXCL3* were highly expressed in cMo-1s and promoted adipogenesis and the inflammatory response (Fig. 5h and Supplementary Fig. 7i), and the levels of the corresponding proteins were increased in the plasma of CD patients who underwent surgery for small intestine stenosis and/or obstruction with CF hyperplasia (Fig. 5i). cMo-1s were more frequent in CF and positively correlated with MCFI, SES-CD and so on (Supplementary Fig. 7j). The metabolic activity analysis demonstrated that cMo-1s were likely to maintain an activated status in

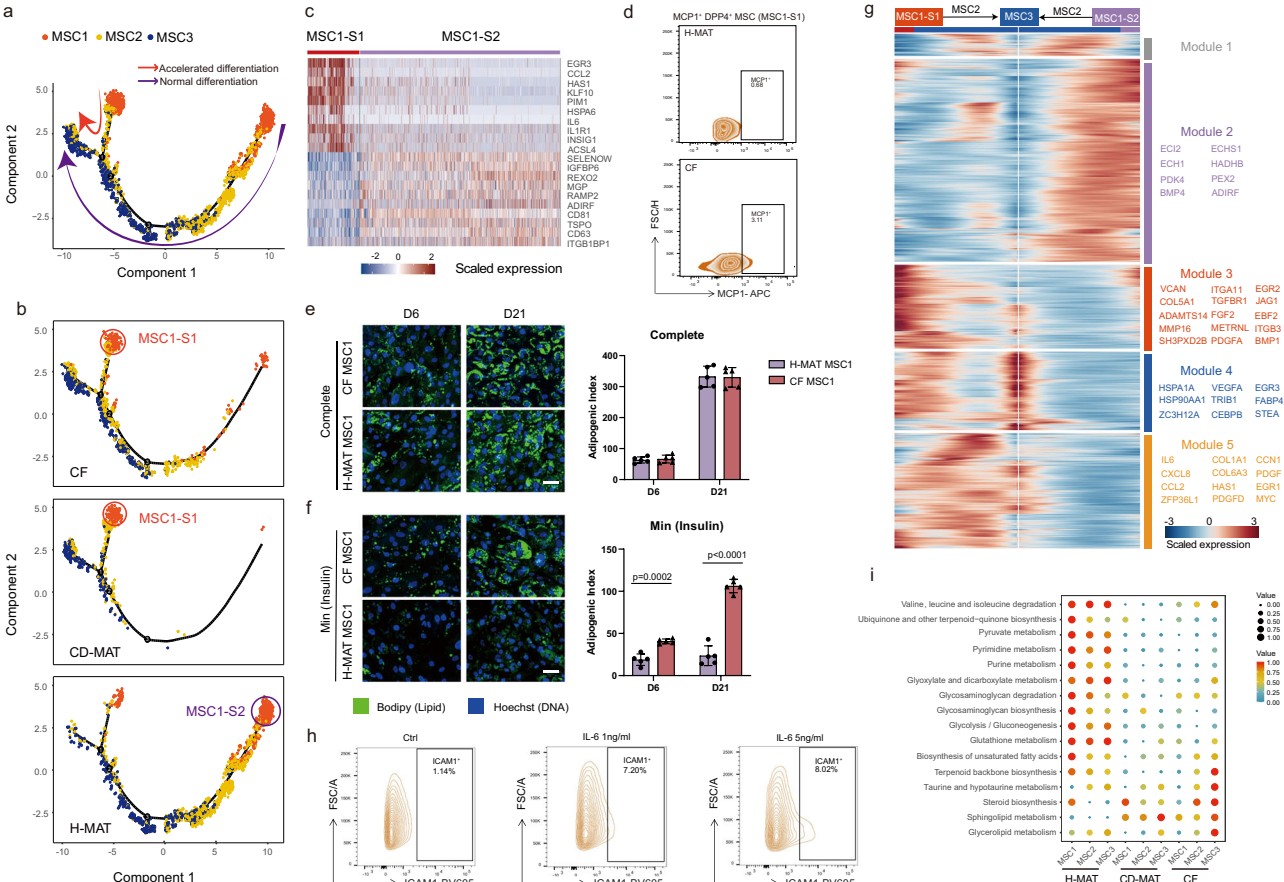

**Fig. 4 | MSC subpopulations expedite adipogenesis in CF. a** Monocle analysis of MSC subpopulations with gene expression profiles indicating accelerated MSC1 differentiation (red arrow) and normal differentiation (purple arrow) in the developmental trajectory. Each point corresponds to a single cell. Cluster information is shown. **b** Monocle analysis of the MSC subpopulations showing tissue source classification as H-MAT (bottom), CD-MAT (middle) or CF (top). **c** Heatmap (red-to-blue) showing the expression of specific marker genes for two MSC1 subsets. **d** Representative flow cytometry zebra plot of monocyte chemoattractant protein 1 (MCP-1, also known as CCL2) expression on MSC1 cells in H-MAT and CF. **e**, **f** Microscopy images of CF-derived and H-MAT-derived MSC1 after exposure to the complete adipogenic differentiation cocktail (**e**) or insulin only (Min) (**f**) for

6 days or 21 days of differentiation (representative of 5 BRs per condition). Scale bars, 50 µM. The data are presented as means ± s.d. and *p* value were determined via two-sided unpaired t-test. **g** Heatmap with spline curves fitted to genes differentially expressed across MSC1-S2 to MSC3 (right arrow) and MSC1-S1 to MSC3 (left arrow) pseudotemporal trajectories, grouped by hierarchical clustering (k = 5). Gene co-expression modules (colour) and exemplar genes from main modules are labelled (right). **h** Flow cytometry showing ICAM1 expression in MSC1 cells after stimulation with IL-6 for 12 h. **i** Metabolic activity analysis of MSC subpopulations from H-MAT, CD-MAT and CF. The circle size and colour darkness both represent the scaled metabolic score. Source data are provided as a Source data file.

CD patients, which could be linked to their unique functions in CF formation (Fig. 5j). The arginine, nicotinate and nicotinamide metabolism pathways were the most upregulated in cMo-1s (Supplementary Fig. 6) and were associated with CD activity[21]. The products of arginine metabolism could facilitate MSC adipogenesis[20].

To investigate how cMo-1s regulate adipogenesis and identify tractable therapeutic targets, the ligand–receptor analysis was performed, which demonstrated that cMo-1s might promote pro-inflammatory and pro-adipogenic mediator expression in MSC1-S1s (Supplementary Data 20). Correspondingly, cMo-1s expressed higher levels of epidermal growth factor receptor ligands known to regulate MSC activation[26]. Interestingly, the results suggested that both cMo-1s and MSC1-S1s in CF were major producers of adipokines and cytokines such as IL-6, but the IL-6 receptor differed in these subsets (IL-6R and HRH1, respectively) (Fig. 5k). By using flow cytometry, we showed that the treatment with conditioned medium from peripheral blood mononuclear cell-derived cMo-1s from CD patients (five males, one female) with CF hyperplasia led to a marked upregulation of ICAM1 expression and adipogenic specificity in MSC1 cells (Fig. 5l). Furthermore, we demonstrated that the effect of cMo-1 cell-conditioned medium on MSC1 cells can be partially blocked by an IL-6 neutralizing

antibody, suggesting the involvement of IL-6 signalling in this process. Altogether, our results indicated that the consequences of cMo-1 accumulation could benefit MSC adipogenesis.

Further pseudotime analyses suggested that a bifurcation of Mac1 and Mac2 fates occurred from cMos but not from ncMos (Supplementary Fig. 8a–c). Analysis of DEGs between Mac1 and Mac2 fates showed a clear bifurcation in gene expression programmes (Supplementary Fig. 8d). These data implicated pro-inflammatory cytokines (IL-1β and IL-6) as significant mediators of the transition from cMos to Mac1s, whereas anti-inflammatory cytokines (IL-1RA) were suggested to regulate the transition to Mac2s (Supplementary Fig. 8e–g). Among myeloid cells, only Mac2s were negatively correlated with CDAI and MCFI (Supplementary Fig. 8h).

### ECs and PCs are enriched and facilitate MSC1-S1 adipogenesis in CF

MAT plasticity and expansion are related to angiogenic capacity[27], which involves EC proliferation and new tube formation that are associated with the initiation and perpetuation of CD[28]. Clustering analysis showed four subpopulations of human MAT ECs: capillary/EC1, venous plexus/EC2, arteries/EC3 and lymphatics/EC4

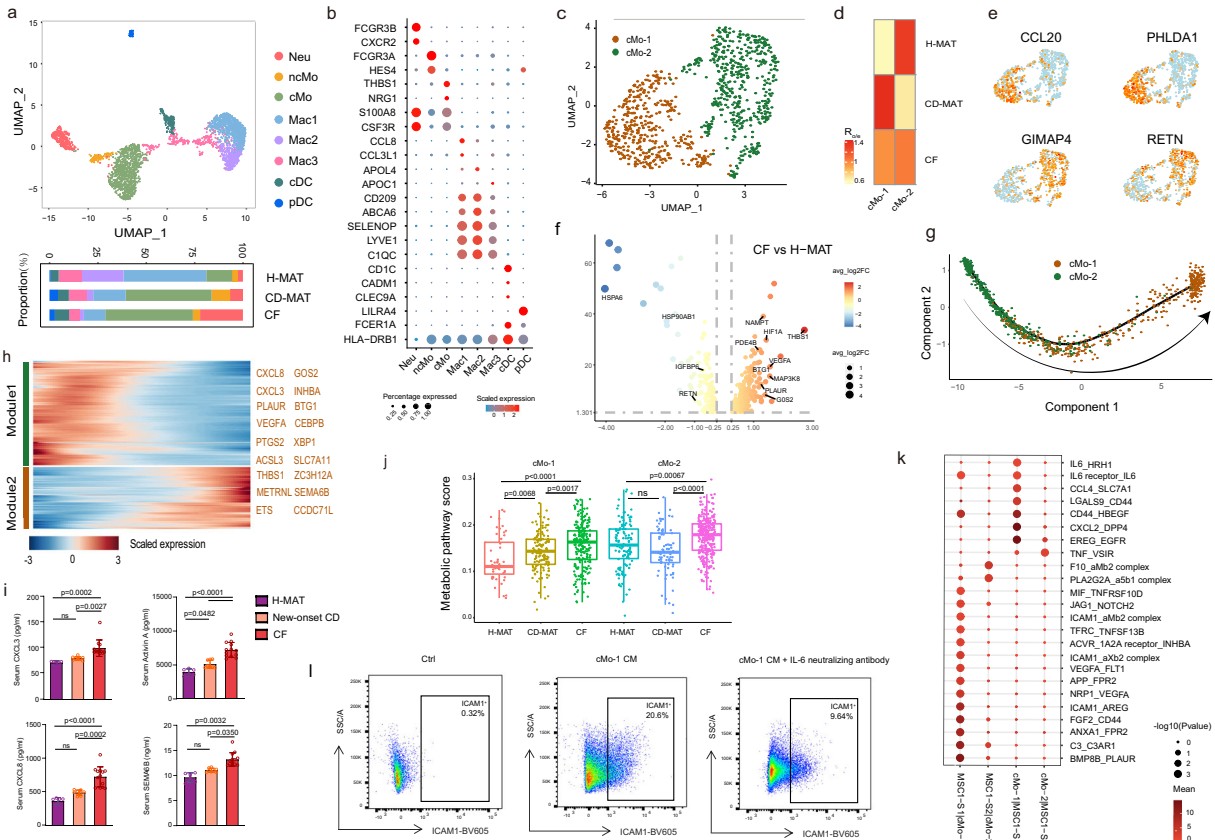

**Fig. 5 | A distinct monocyte subset accumulates in CF and is pro-inflammatory and pro-adipogenic. a** scRNA-seq-generated UMAP plot of myeloid cells distinguishing eight individual subpopulations. Coloured bars indicate the proportion of myeloid cells in H-MAT, CD-MAT and CF. **b** Dot plot showing selected top differentially expressed marker genes for eight subpopulations of myeloid cells. Colour saturation indicates the strength of expression in positive cells, while dot size reflects the percentage of each cell cluster expressing the gene. **c** UMAP plot of monocytes distinguishing two individual subsets. **d** The proportion of cMo1 and cMo2 of H-MAT, CD-MAT and CF estimated by Ro/e score. **e** Expression levels of representative marker genes for cMo-1 and cMo-2 are plotted onto the UMAP. **f** The volcano plot represents the differentially expressed genes of cMo-1 between CF and H-MAT. **g** Monocle analysis of cMo cells with gene expression profiles indicating pseudotime directionality (arrow). Each point corresponds to a single cell. Cluster information is shown. **h** The differentially expressed genes (rows) along the pseudotime (columns) are clustered hierarchically into two profiles. Heatmap showing the expression of representative identified genes. Gene co-expression modules

(colour) and exemplar genes from module 2 are labelled. **i** ELISA for CXCL3, Activin A, CXCL8 and SEMA6B detection in serum obtained from blood samples taken from H-MAT ($n = 6$), new-onset CD ($n = 8$) or CF ($n = 13$, from CD patients who underwent surgery for small intestine stenosis and/or obstruction with varying degrees of CF hyperplasia). The data representing mean ± s.d. and $p$ value were generated using one-way ANOVA. **j** Boxplots comparing the metabolic activity scores of cMo-1 and cMo-2 in H-MAT ($n = 2$), CD-MAT ($n = 3$) and CF ($n = 6$). The centre line represents the median, box hinges represent frst and third quartiles and whiskers represent ± 1.5x interquartile range. **k** Bubble plots showing ligand–receptor pairs between cMo and MSC1 cells in CD patients and control subjects. The circle size represents the log-normalized $P$ value, whereas the colour represents the log-transformed mean expression of ligand and receptor. **l** Flow cytometry showing ICAM1 expression in MSC1 cells after co-cultured with cMo-1 culture medium with or without IL-6 neutralizing antibody for 12 h. **j, k** Data were analysed using two-sided Wilcoxon rank sum test. Source data are provided as a Source data file.

(Fig. 6a, b). EC1s were identified by *CA4* and *RGCC* (Fig. 6b, Supplementary Fig. 9a and Supplementary Data 21). The top venous marker was *ACKR1* in EC2s. EC3s were characterized by *GJA5* and *FBLN5*. EC4s were identified by the canonical lymphatic markers *PROX1* and *PDPN*. To fully annotate ECs, we determined functional expression profiles (Supplementary Fig. 9b) and performed TF regulon analysis (Supplementary Fig. 9c).

Our scRNA-seq data revealed that EC1s were the most abundant in CF (Fig. 6c), which was confirmed immunohistochemically by protein-level validation of CA4 (Fig. 6d). EC1 DEGs revealed upregulated genes related to angiogenesis and leucocyte transendothelial migration (Fig. 6e, Supplementary Fig. 9d and Supplementary Data 9). Further clustering identified two subsets of EC1s (Fig. 6f). *CXCL12* expression distinguished subset 1 of EC1 (EC1-S1) (Fig. 6g and Supplementary Fig. 9e). Notably, CXCL12 is known to promote tube formation and is a potential novel target for tumour angiogenesis[29]. During the transition from EC1-S2s to CF-enriched EC1-S1s, genes involved in vasculature development, such as *VEGFC*[28], gradually increased during CF formation

(Fig. 6h and Supplementary Fig. 9f, g). EC1-S1s highly expressed pro-angiogenic TFs (*SOX4*, *MYC* and *JUNB*) during this transition (Fig. 6i). Ligand–receptor analysis between EC1 subsets and other cell groups demonstrated that MSC1-S1s and cMo-1s may preferentially induce EC1-S1 chemotaxis and angiogenesis (Fig. 6j). For example, RPS19-C5AR1 could promote new blood vessel formation[30], providing clues for targeted therapy against angiogenesis in the adipogenic niche.

PCs can interact with ECs to support vasculature stabilization[31]. PCs covered endothelial abluminal surfaces and increased in CF (Fig. 6k). Clustering of PCs revealed three subpopulations: PC1 and PC2 highly expressed *MYH11* and *ACTG2*, whereas PC3 expressed *STEAP4* (Supplementary Fig. 10a–c). We identified differentially expressed markers, TFs and functions (Supplementary Fig. 10d–f and Supplementary Data 22). We performed pseudotime analyses, which suggested a developmental trajectory from PC3 to PC2 and then to PC1 (Supplementary Fig. 10g–i). Combined, these results suggest that ECs, assisted by PCs, might require activation by mediators secreted by cMo-1 and/or MSC-S1s to exert proangiogenic effects during CF formation.

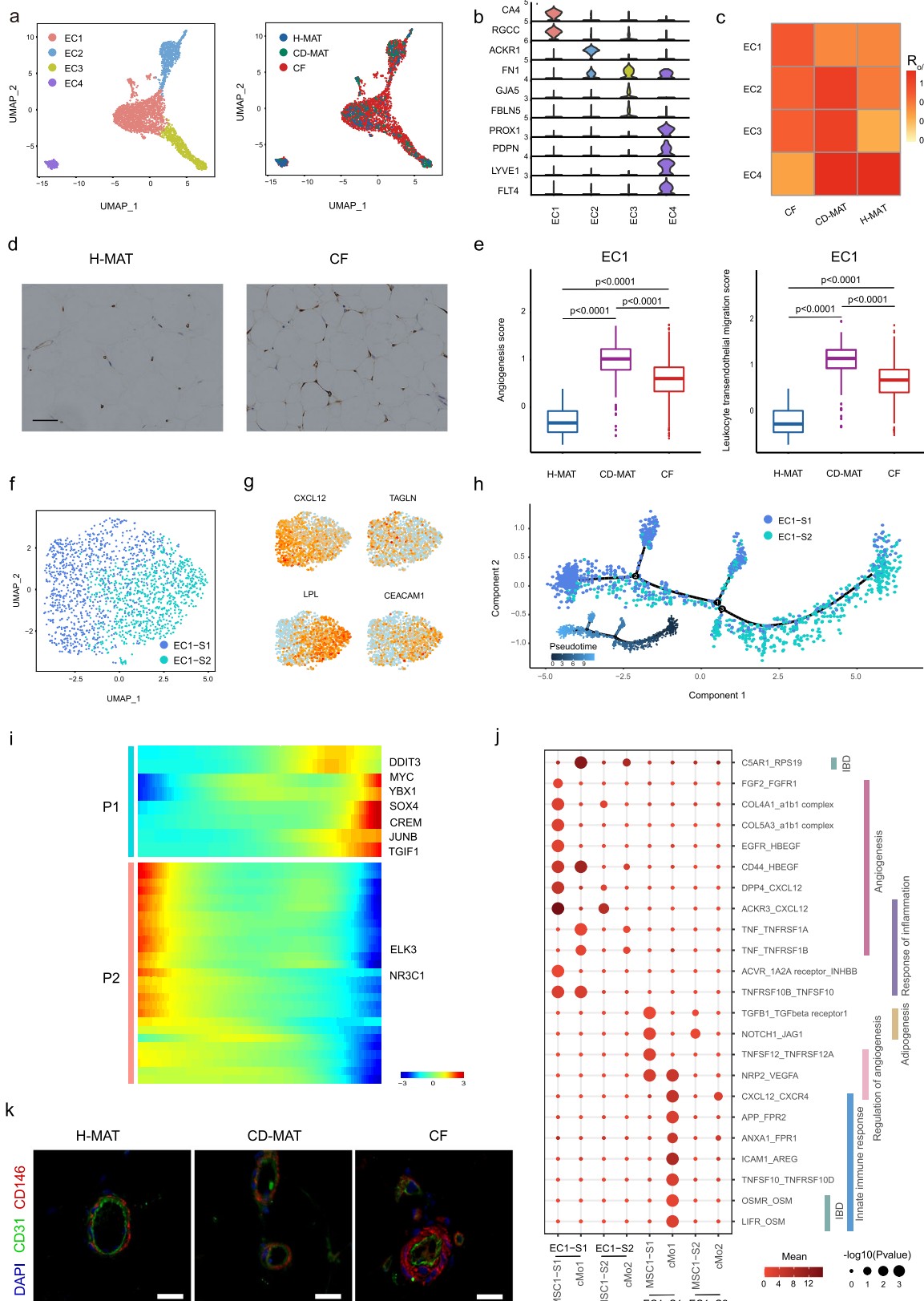

## Discussion

Creeping fat is a characteristic feature in CD patients and correlates with intestinal fibrosis and stricture complications[32,33]. However, recent studies have suggested a protective role for CF in CD in terms of an enveloping barrier with the potential to restrict intestinal inflammation[3]. Whether CF in human CD is harmful or beneficial is a longstanding question. By the time of surgical resection, fibrotic and hyperplastic mesenteric adipose encases the underlying ileum which is also significantly fibrotic. The findings suggest that preventing or eliminating CF expansion may be a novel tactic for improving the efficacy and complications of CD and that elucidating its mechanisms may elicit new modes of treatment. Therefore, we comprehensively

**Fig. 6 | EC subpopulations inhabit the adipogenic niche. a** scRNA-seq-generated UMAP plots of ECs distinguishing four individual subpopulations (left) and tissue sources (right). **b** Violin plots displaying the expression of representative well-known marker genes across four EC subpopulations. The y-axis represents log-scaled normalized UMI counts. **c** The proportion of four EC subpopulations of H-MAT, CD-MAT and CF estimated by Ro/e score. **d** Immunohistochemical staining of CA4 expression in the H-MAT and CF vasculature (representative of 5 BRs per group). Scale bars, 25 μM. **e** Boxplots comparing the scores of angiogenesis and leucocyte transendothelial migration in EC1 of H-MAT (*n* = 2), CD-MAT (*n* = 3) and CF (*n* = 6). The centre line represents the median, box hinges represent frst and third quartiles and whiskers represent ± 1.5x interquartile range. **f** UMAP plot of EC1 distinguishing two individual subsets. **g** Expression levels of representative marker genes for EC1-S1 and EC1-S2 are plotted onto the UMAP. **h** Monocle analysis of the EC1 subsets indicating pseudotime directionality (bottom left) and cell type; EC1-S1 (blue) and EC1-S2 (cyan). **i** The differentially expressed genes (rows) along the pseudotime (columns) are clustered hierarchically into two profiles. Heatmap showing the expression of representative TFs in two profiles across single cells. **j** Bubble plots showing ligand–receptor pairs between EC1s and other cell groups in CD patients and control subjects. The circle size represents the log-normalized *P* value, whereas the colour represents the log-transformed mean expression of ligand and receptor. **k** Immunofluorescence confirmed the existence and distribution of pericytes in H-MAT, CD-MAT and CF (representative of 5 BRs per group). Scale bars, 50 μM. **e**, **j** Data were analysed using two-sided Wilcoxon rank sum test. Source data are provided as a Source data file.

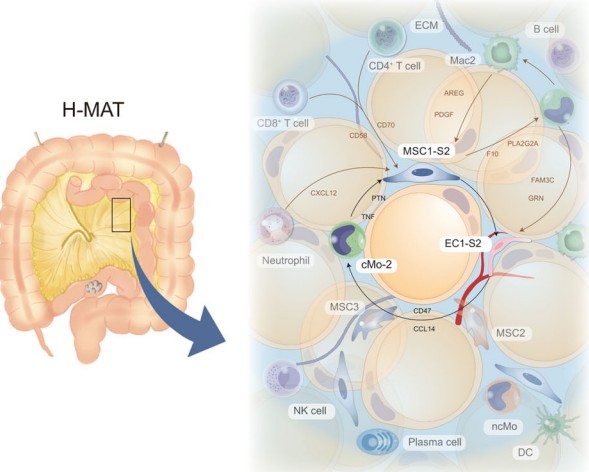

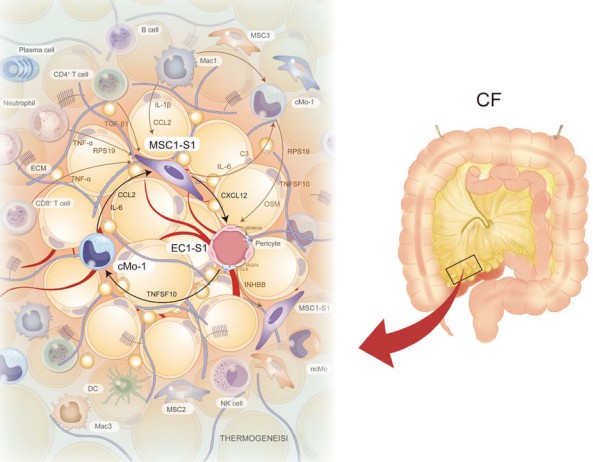

**Fig. 7 | Predicted regulatory network centred on MSC1-S1 in the adipogenic niche of CF.** Increased IL-6, CCL2 and INHBA from cMo1 could drive the differentiation from MSC1-S1 to MSC2, and promote hyper-inflammatory and pro-adipogenic mediator expression in MSC1-S1 conversely expediting adipogenesis of MSC1-S1. Accumulated CXCL2 production from MSC1-S1 coupled with RPS19, TNFSF10 and OSM expression from cMo1 could induce EC1-S1 proliferation and angiogenesis, then facilitating new blood vessels which emerged as a critical energy provider for CF formation and simultaneously attracted cMo-1 to the adipogenic niche.

investigated the potential mechanisms underlying CF formation by utilizing multi-omics and an ex vivo primary cell interaction system. We identified CCL2⁺DPP4⁺ MSCs, CCL20⁺CD14⁺ monocytes and IL-6 as central cues for CF development (Fig. 7).

Here, we found adipocyte shrinkage and nascent adipocyte emergence in CF, concurrently with immune cell infiltration, ECM deposition and poly-partitioned lipid droplets surrounding. To date, lipidomics analysis has not been reported, but our results revealed an insufficient lipid metabolic status of CF, which reflects, to a large extent, the distinct forms of lipid storage and lipolysis in adipocytes and may be tightly associated with the marked increase in the populations of smaller and nascent adipocytes in CF. We showed that upregulation of multiple pro-adipogenic-related lipid metabolic pathways distinguishes CF. Core features include sphingolipid and glycerolipid metabolism, which are positively associated with CDAI and C-reaction protein[34,35]. This suggests that distinct forms of lipid metabolism in CF could aggravate intestinal inflammation, perhaps even fibrosis, when the opportunity arises.

Directly relevant to the formation of CF is which cellular composition and phenotype most effectively induces fat cell differentiation. We next comprehensively depicted the single-cell atlas of the adipogenic niche in human MAT and the developmental hierarchy of MSCs in de novo formation of CF. CF tissues were distinguished by ECM production and stromal populations that were nearly absent from H-MAT and characterized as both pro-fibrotic and pro-adipogenic. We identified and functionally assessed three MSC subpopulations in

human MAT: MSC1 displayed higher proliferative activity and lower adipogenic capacity, but decreased in CF; MSC2 and MSC3, which were dramatically increased in CF, were highly adipogenic and relatively restricted to the adipocyte lineage. One of the striking findings in single-cell analyses was the definition of a developmental hierarchy of MSCs that are active during early human MAT adipogenesis. Collectively, a scenario emerges in which a developmental shift exists in CF: MSC1s significantly decrease and expedite adipogenic differentiation into MSC2s and MSC3s, contributing to dystrophic adipocyte accumulation during CF formation.

To our knowledge, this is the first report proposing that MSC1s have two transcriptional states along MSC differentiation trajectories. We convincingly demonstrate a previously unreported pathogenic subset, CCL2⁺DPP4⁺ MSCs (MSC1-S1s), which constituted up to 67–99% of MSC1s in CF but only ~4% in H-MAT. In our research, MSC1-S1s had unique pathogenic characteristics, including accelerated differentiation into MSC2s and MSC3s (also known as committed preadipocytes) and then dystrophic adipocytes, and this process was mainly responsible for CF formation in CD. By using scMetabolism, we found that the three subpopulations of MSCs were less metabolically active, interestingly, MSC1-S1s displayed the highest lipid metabolic activity, which might contribute to pathologic behaviours in CF. For example, sphingolipid, arginine and nicotinamide metabolism were upregulated in MSC1-S1 cells, which is essential for MSC adipogenic differentiation and positively associated with active CD progression and unfavourable prognosis[36].

We therefore were particularly intrigued by which cell types and/or molecules could trigger MSC1 towards a phenotype that is able to expedite adipogenesis. Gene expression of the transition from MSC1-S1s to MSC2s and MSC3s exhibited upregulation of genes involved in the cellular response to inflammation. We found that the frequency of cMo-1s was greater than 20% among all innate immune cells in CF but limited to ~3% in H-MAT, and it was positively correlated with CDAI and MCFI. Ligand–receptor analysis also revealed that cMo-1s had a unique effect on MSC1-S1s, which confirmed in vitro experiments. A discrepancy in the impacts of cMo-1s and conventional monocytes on MSCs was described by Svensson et al.[37], which may be attributed to the diverse functions of various monocyte subsets in disease states. MSCs from ankylosing spondylitis augmented monocyte migration and increased macrophage polarization, which indicated that MSCs act on monocytes in return, but further investigation is needed. Preclinical research revealed that targeted monocytes and inflammatory macrophages that specifically express carboxylesterase 1 could achieve therapeutic benefit in IBD[38], however, whether they could prevent intestinal fibrosis and CF formation was not reported.

Given that a single molecule, IL-6, can mediate epigenetic reprogramming of gut stem cells to acquire enduring epigenetic memories, a phenomenon that contributes to both enhanced immunity and predisposition to inflammatory disorders, such as CD, and worsened pathology in a model of colitis[39], we confirmed that IL-6 stimulates MSC1s towards a pro-adipogenic phenotype that leads to CF formation and penetrating inflammatory fibrosis in patients with CD. A recent very elegant investigation suggested that gp130 (also known as IL-6ST) blockade may benefit some CD patients with NOD2-driven fibrosis, potentially as a complement to anti-TNF therapy[40].

Lack of in vivo experimental models of CF is a serious limitation. Although many scholars have made numerous attempts to develop a model[41], there is still no universally acknowledged model for developing CF. In addition, factors other than components in the adipogenic niche that were the focus of this investigation, such as mediators from the gut and lymph vessels, may be capable of activating MSC1s towards a pro-adipogenic phenotype[42]. Besides, the analysis was based on pooled samples of both males and females and disaggregated analysis was not performed due to limited sample size and accessibility of patient samples, and further work would be needed to see to what extent these findings are dependent on sex.

A better understanding of pathogenic MSC1-S1 infiltration and its potent ability to accelerate adipogenesis has been achieved by the description of the unique CF formation-specific signals we found in CD-MAT and CF of CD patients, which has not been reported previously and is likely to provide fundamental clues regarding the pathogenic mechanisms and/or disease progression in CD. Given the dynamic changes in MSC subpopulations and unique disease-specific MSC1 subset functions in CF, future studies that include inhibitor and/or blocker screening against expediting adipogenic targets on MSC1-S1s will be important.

## Methods

### Patients and specimen preparation

Patients were recruited at Nanfang Hospital and the Sixth Affiliated Hospital of Sun Yat-sen University (Guangzhou, China). This study was performed following the ethical guidelines of the Declaration of Helsinki and was approved by the local ethical committee (NFEC-2021-053). Informed consent was obtained from all enrolled patients before collection of specimens and clinical information. All patients approved publication of information that identifies individuals, received both oral and written project information, and signed written consent. China's Ministry of Science and Technology has approved the export of genetic information and materials pertaining to this research (2023BAT1107).

Ileum-attached CF and adjacent uninvolved mesenteric adipose tissue (CD-MAT) and blood were obtained from 60 patients undergoing surgical resection due to complications from CD. The diagnosis of CD was established using typical criteria[1]. The hyperplastic mesenteric adipose tissue outside the inflamed and stenotic ileum was taken as CF, and CD-MAT was taken from the uninvolved MAT that does not undergo morphological alterations in response to the inflammatory process in CD. We also collected healthy ileum-attached MAT (H-MAT) and blood from control subjects undergoing resection in non-inflammatory bowel disease ileal surgery. The patients for metabolic analysis were consecutively selected and samples that met our quality control criteria were included. The basic characteristics and demographics of the patient cohort are detailed in Supplementary Data 1a, 1b, 3 and 6. This included details such as age, sex, BMI, disease duration, disease location, disease behaviour, and medication use.

### Isolation and culture of stromal vascular fraction (SVF) cells from human MAT

The tissue was incubated in Hank's balanced salt solution (HBSS) with 2500 U potassium penicillin, 2500 μg of streptomycin sulfate, 625 μg of amphotericin B (PSF, Lonza, Basel, Switzerland) for 3 h. Then the mesenteric tissue was cut into small pieces (<1 mm) and incubated in digestion buffer [1.5 mg/mL collagenase I, 1 mg/mL collagenase II and 1% penicillin–streptomycin in phosphate-buffered saline (PBS) with 1% foetal bovine serum (FBS), all from SIGMA] for 1 h at 37 °C under agitation. The digestion was quenched with DMEM containing 10% FBS, and the dissociated cells were passed through a 100-μM filter (FALCON) and then subjected to centrifugation at $400 \times g$ for 5 min at room temperature (RT). The cell pellet containing the SVF was resuspended in red blood cell lysis buffer for 5 min at RT and then quenched in DMEM containing 10% FBS. After centrifugation, cells were resuspended in HBSS containing 3% FBS and kept on ice for the duration of processing. Cell concentration and viability were determined on a Cellometer Auto 2000 (Nexcelom, USA) after AO/PI staining, and cells with viability of greater than 80% were subjected to 10× Genomics scRNA-seq.

### Flow cytometry analysis

The details of the antibodies were as follows: human CD31 (diluted 1:100; PE; BD Biosciences, cat: 303106), CD45 (diluted 1:100; FITC; BD Biosciences, cat: 555482), CD26 (diluted 1:100; R718; BD Biosciences, cat: 752183), CD54 (diluted 1:100; BV605; BD Biosciences, cat: 740404), CD142 (diluted 1:100; BV421; BD Biosciences, cat: 744003) and CD146 (diluted 1:100; BV510; BioLegend, cat: 361021) and fixable viability stain 780 (diluted 1:500; APC/cy7; BD Biosciences, cat: 565388). For testing nuclear markers in MSC1-S1s, cells were resuspended in 1 mL fixation/permeabilization working solution (Invitrogen; cat: 00-5523-00). After 30 min in the dark, 2 mL of 1X permeabilization buffer was added and then subjected to centrifugation at $400 \times g$ at RT for 5 min. Subsequently, the cells were stained with MCP-1 (diluted 1:100; APC; BioLegend, cat: 502612) for 30 min in the dark and washed with 2 mL of 1× permeabilization buffer.

The following gating strategy was applied to analyse the cells: doublets and dead cells were excluded based on forwards and side scatter. Immune cells were purified using CD45-FITC and ECs were identified using CD31-PE. Gates were created to collect MSCs (CD31⁻CD45⁻CD146⁻) and PCs (CD31⁻CD45⁻CD146⁺). Then MSC1s (CD26⁺CD142⁻CD54⁻), MSC2s (CD26⁻CD142⁺CD54⁻) and MSC3s (CD26⁻CD142⁻CD54⁺) cells were identified from MSCs. MSC1-S1s were identified from MSC1s by staining with MCP-1-APC.

MSCs were identified using a Human MSC Analysis Kit (BD Biosciences, cat: 562245) according to the manufacturer's instructions. Isolated cells were >99% positive for CD90, CD105, CD73 and CD44 but negative for CD34, CD11b, CD19, CD45, CD14, CD79α or HLA-DR, characterizing them as MSCs.

All flow cytometric experiments were performed by BD FACSAria (BD Biosystems). Flow cytometry data were analysed by Flow Jo (Ashland, OR).

## MSC culture and differentiation

MSCs were plated on a cell culture dish and cultured in a specific MSC culture medium supplemented with 10% FBS and 2% penicillin–streptomycin (Cyagen Biosciences Inc). The cells were incubated for 24 to 48 h to facilitate attachment before the induction of adipogenic, osteogenic or chondrogenic differentiation. Adipogenic differentiation was carried out with the human adipose-derived stem cell adipogenic differentiation kit (Cyagen Biosciences Inc, GUXMX-90031) for 21 days according to the manufacturer's protocol. Osteogenic differentiation was performed by using the MesenCult Osteogenic Differentiation Kit (Stem Cell Technologies cat: 05465). The medium was changed every 3 days until bone matrix formation occurred (10–15 days). Chondrogenic differentiation was performed by using the MesenCult Chondrogenic Differentiation Kit (Stem Cell Technologies, Cat: 05455). The medium was replaced with 0.5 mL medium every 3 days for a total of 21 days.

## Quantification of the adipogenic index

To assess adipogenesis, lipid droplets were labelled with BODIPY 493/503 (Invitrogen cat: D3922) and nuclei were labelled with Hoechst 33342 (12.3 mg/ml; Thermo Fisher cat: 62249). Images were taken by ZEISS LSM880. The size and number of BODIPY-positive lipid droplets were calculated by ImageJ as previously described[15]. In brief, nuclei (blue channel) were quantified by applying Gaussian blur (3 Sigma), thresholding, watershed segmentation, and counting. Lipid accumulation was quantified by applying Gaussian blur (2 Sigma), thresholding, and quantification of the total area above the threshold. The adipogenic index of each well was assessed by dividing the total lipid area by the total number of nuclei. For each sample at least 10 fields were analysed.

## Cell proliferation assay

Cell proliferation was determined by Cell Counting Kit-8 (CCK-8, DOJINDO, cat: CK04) reagent according to the manufacturer's instructions. Cells were seeded in 96-well plates and allowed to attach for 24 h with 5 replicates for each time point (0−48 h). Then the cells were incubated with 10 μL CCK-8 for 1 h and the number of cells was counted with a SpectraMax M4 (Molecular Devices) under absorbance at A450.

## Gene expression analysis by quantitative reverse transcription PCR

Total RNA was prepared from cells using TRIzol reagent (Takara) in accordance with the manufacturer's instructions. First-strand cDNA synthesis was carried out by reverse transcription using PrimeScript™ RT Master Mix (Takara), and qPCR was performed using SYBR Green Master Mix (Takara). Reactions were performed as described previously using the LightCycler 480® System (Roche, Basel, Switzerland)[43]. The specific sequences of the primer pairs are provided in Supplementary Data 13. The mRNA levels of the target genes were determined using the $2^{-\Delta\Delta CT}$ method after normalization to cyclophilin A. Sequences of primers are provided in Supplementary Data 15.

## IL-6, CXCL2 and CXCL8 stimulation

Purified MSC1s were seeded at 80,000 cells per well in 6-well plates in specific MSC culture medium for 2−3 days. Then, MSC1s were starved in 0.2% FBS MSC culture medium for 8 h before IL-6 (1 ng/mL or 5 ng/mL), CXCL2 (1 ng/mL or 5 ng/mL) or CXCL8 (1 ng/mL or 5 ng/mL) stimulation. After 12 h of stimulation, MSCs were stained with CD26-R718, CD54-BV605, CD142-BV421 and Fixable Viability Stain 780-APC-cy7. Changes in cell surface biomarkers were detected by flow cytometry.

## Magnetic-activated cell sorting

Immune cells and ECs were excluded from the SVF by FcR Blocking Reagent, CD31-Microbeads and CD45-Microbeads (Miltenyi Biotec, cat: 130-091-935, 130-045-801). Subsequently, the selected cells were incubated with anti-CD142 (Miltenyi Biotec, cat: 130-115-682), anti-CD54 (Biolegend, cat: 322706) or anti-CD26 (Biolegend, cat: 302718) antibodies for 15 min, then washed with running buffer (Miltenyi Biotec, cat: 130-121-565) and incubated with anti-biotin microbeads (Miltenyi Biotec, cat: 130-090-485) for 15 min. A concentration of $1 \times 10^8$ cells per millilitre were incubated with 5 μg of each biotin-conjugated antibody. Finally, MSC1s (CD26⁺CD142⁻CD54⁻), MSC2s (CD26⁻CD142⁺CD54⁻) and MSC3s (CD26⁻CD142⁻CD54⁺) cells were magnetically enriched following the manufacturer's instructions.

Peripheral blood mononuclear cells (PBMCs) were isolated by Ficoll-Paque (1.077 g/mL; Invitrogen) density gradient centrifugation. After incubation for 2 h or overnight the cells were gently agitated, and the adherent cells were collected. Monocytes (more than 85% CD14⁺) were purified from adherent cells by using the MACS Pan Monocyte Isolation Kit (Miltenyi Biotec, cat: 130-096-537). cMo1s stained with the anti-CD98 antibody also known as SLC7A5 (Miltenyi Biotec, cat: 130105550) were subsequently purified subsequently, and $1 \times 10^6$ cells per millilitre were cultured in 3 mL RPMI 1640 with 10% FBS. A concentration of $1 \times 10^8$ cells per millilitre were incubated with 5 μg of each biotin-conjugated antibody.

## Co-culture of MSC1s and cMo1-conditioned medium

Purified MSC1s were seeded at 80,000 cells per well in specific MSC culture medium for 3 days. LPS (10 ng/mL)-exposed cMo1s were washed and cultured in RPMI 1640 with 2% FBS for 24 h before co-culture with MSC1s, whereas MSC1s were starved in MSC culture medium with 0.2% FBS for 12 h. Then MSC1s were cultured in cMo1-conditioned medium with or without IL-6 neutralizing antibody (7.5 μg/mL). After 12 h of co-culture, the cells were stained with CD45-FITC, CD26-R718, CD54-BV605, CD142-BV421 and the fixable viability stain 780-APC-cy7. Finally, the changes in cell surface biomarkers were detected by flow cytometry.

## Immunohistochemistry and immunofluorescence

Paraffin-embedded MAT sections were cut to a thickness of 5 μM, deparaffinized in xylene, and rehydrated in a graded alcohol series. Antigen retrieval was achieved by microwaving at 100 °C with an antigen unmasking solution (10 mM citrate buffer, pH 6.0). Hydrogen peroxide (3%) was used to eliminate endogenous peroxidase. After serum blocking in serum, the sections were subsequently incubated overnight at 4 °C with primary antibodies against CD31 (1 μg/ml; Mouse monoclonal; Abcam, cat: ab9498), CD26 (diluted 1:200; Rabbit monoclonal; DPP4, Abcam, cat: ab215711), CD142 (diluted 1:100; Rabbit monoclonal; F3, Abcam, cat: ab228968), CD54 (1000 μg/ml; Mouse monoclonal; ICAM1, Proteintech, cat: 60299) and CD146 (diluted 1:200; Rabbit monoclonal; Abcam, cat: ab75769) followed by incubation with biotinylated secondary antibodies (Zhongshanjinqiao, Beijing, China). Immunostaining was examined with an Olympus BX-53 microscope (Olympus). The secondary antibodies used in this study were as follows: Goat Anti-Rabbit IgG (HRP) (diluted 1:4000, Abcam, cat: ab205718), Alexa Fluor 594 Donkey Anti-Mouse IgG (H + L) (diluted 1:400, Life Technologies, cat: A21203), and Alexa Fluor 647 Donkey Anti-Rabbit IgG (H + L) (diluted 1:200, Abcam, cat: ab150075).

For CA4 immunohistochemistry, all washes were carried out with TBST (dH₂O, 200 mM Tris, 1.5 M NaCl, 1% Tween-20 (all Sigma–Aldrich) pH 7.5) and peroxidase blocking was carried out for 30 min in 0.6% hydrogen peroxide in methanol. Sections were incubated with CA4 (diluted 1:200; Rabbit Polyclonal; Proteintech, cat: 13931) for 20 min

and washed in TBST before a haematoxylin (Vector Laboratories, SK-4285) counterstaining.

## Scanning electron microscopy

A sharp blade was used to cut and harvest fresh MAT blocks quickly within 1–3 min. The washed tissue blocks were immediately fixed with electron microscopy fixative for 2 h at RT and then washed with PBS 3 times for 15 min each. Then, the tissue blocks were incubated with 1% $OsO_4$ (Ted Pella Inc.) in PBS for 1–2 h at RT. Afterwards, the tissue was washed in PBS 3 times for 15 min each and then dehydrated before being dried with a critical point dryer. Specimens were attached to metallic stubs using carbon stickers and sputter-coated with gold for 30 s. Finally, images were obtained with a scanning electron microscope. Nascent adipocytes (diameter < 10 μM) were quantified in five fields. All evaluations were performed in a blinded manner.

## Quantitative lipid profiles

**Extraction and analysis of lipid metabolites.** The whole-MAT samples were thawed on ice. One millilitre of lipid extraction solution, 20 mg MAT samples and steel balls were added to a sample tube and homogenized. Then, the steel balls were removed and mixed by vortexing for 2 min. The mixed samples were emulsified by sonication for 5 min, mixed with 500 μL water, vortexed for 1 min, and then centrifuged at 4 °C and 12,000 × $g$ for 10 min. The 500 μL supernatant was collected and dried with nitrogen and redissolved in 100 μL of mobile phase B. After vortexing for 1 min, the sample was centrifuged at 14,000 × $g$ for 15 min at 4 °C, and UPLC–MS/MS analysis was carried out.

**Liquid chromatography and mass spectrometry.** Ultra-performance liquid chromatography (UPLC, ExionLC™ AD, https://sciex.com.cn/) and tandem mass spectrometry (MS/MS, QTRAP® 6500+, https://sciex.com.cn/) were the main instrument systems for data acquisition. The chromatographic columns from Thermo Accucore™ C30 (2.6 μM, 2.1 mm × 100 mm i.d.) were used. The solvent system was as follows: A, acetonitrile/water (60/40, V/V, 0.1% formic acid, 10 mmol/L ammonium formate); B, acetonitrile/isopropanol (10/90, VV/V, 0.1% formic acid, 10 mmol/L ammonium formate). The gradient programme was $t = 0$ min: A/B (80:20, V/V); $t = 2$ min: A/B (70:30, V/V); $t = 4$ min: A/B (40:60, V/V); $t = 9$ min: A/B (15:85, V/V), $t = 14$ min: A/B (10:90, V/V); $t = 15.5$ min: A/B (5:95, V/V); $t = 17.3$ min: A/B (5:95, V/V); $t = 17.5$ min: A/B (80:20, V/V); $t = 20$ min: A/B (80:20, V/V); and the flow rate was 0.35 mL/min, with a temperature of 45 °C. Subsequently, the effluent was alternatively connected to an electrospray ionization (ESI)-triple quadrupole-linear ion trap (QTRAP)-MS. Linear ion trap and triple quadrupole (QQQ) scans were acquired on a triple quadrupole-linear ion trap mass spectrometer (QTRAP), QTRAP® 6500 + LC–MS/MS System, equipped with an ESI Turbo Ion-Spray interface, operating in positive and negative ion mode and controlled by Analyst 1.6.3 software (Sciex, Framingham, USA). Lipid contents were detected by LC–MS/MS (AB Sciex QTRAP 6500, Framingham, USA), the service provided by MetWare, Co.,Ltd (http://www.metware.cn/, Wuhan, China), which lipidomics platform based on an in-house database (MWLDB, v 3.0).

**Qualitative and quantitative analysis.** The MWLDB (Metware lipidomics database v 3.0) was constructed based on standard materials to qualitatively analyse the data detected by mass spectrometry. The multiple reaction monitoring (MRM) mode of triple quadrupole mass spectrometry was applied for the quantification of analytes.

**Bioinformatics and statistical analysis.** The R programming language (version 4.1.1) was applied for statistical analyses and the generation of graphs. The characteristic ions of each lipid metabolite were processed by multiple reaction monitoring based on the Metware database, and then MultiQuant (version 3.0.3, AB Sciex) software was used to analyse the chromatogram review and peak area integration of the off-board mass spectrometry file of the sample. Each chromate graphic peak represents a lipid metabolite, and the area under the peak corresponds to the relative content. The qualitative and quantitative analysis results of all lipid metabolites were calculated based on the linear equation and calculation formula. The variable importance in projection (VIP) was calculated based on orthogonal partial least squares discriminant analysis (OPLS-DA), and the $P$ value and fold change (FC) of the non-parametric test were used in combination to screen the differential lipid metabolites. Then, the cut offs of VIP ≥ 1, FC > 1.5 or FC < 0.66 and $P < 0.05$ were used as standards to screen differential lipid metabolites.

## RNA-seq and data analysis

**10x Genomics Chromium library construction and sequencing.** ScRNA-seq libraries were generated with the Chromium Single Cell 3′ v.3 assay (10× Genomics). Libraries were sequenced using the NovaSeq 6000 platform (Illumina) to a depth of approximately 500 million reads per library with 2 × 150 read length. Raw reads were aligned to the human genome (hg38), and cells were called using cellranger count (v5.0.1).

**Quality control of scRNA-seq data.** We first discarded cells with low quality based on two criteria: contamination rate with mitochondrial genes (<15%), lower and upper limits of unique molecular identifiers in the range of 500 < UMIs < 4000 and lower and upper limits of UMI counts in the range of 2000 < UMI counts < 20,000. Moreover, to remove ambient RNA contamination from droplets, we applied the SoupX R package (v1.5.2) with the default settings, and we also used the DoubletFinder R package (v2.0.3) with the default settings to remove candidate doublets. The quality control and normalization procedures of the scRNA-seq data were conducted with the Seurat R package (v4.0.2).

**Cell clustering and cell-type annotation.** From the filtered cells, the gene expression matrices were normalized to the total UMI counts per cell and transformed to the natural log scale. To correct the batch effects, we integrated different samples using reciprocal PCA (rPCA) implemented in Seurat. We used the FindVariableFeatures function to obtain the top 3000 highly variable genes (HVGs) of each sample and as the input dataset for batch effect correction. Using the FindIntegrationAnchors function with the default dimensions (1:20), we found a set of pairwise correspondences between individual cells. These anchors are used for downstream integration of the objects. We used the IntegrateData function with the previously computed anchor set as a parameter to integrate all sample Seurat objects. The default dimensions parameter (1:20) was used for the anchor-weighting procedure. The integrated dataset on all cells was then used to scale and centre the genes and compute the principal components. After PCA to reduce dimensionality and build k-nearest neighbour graphs (k = 10) of the cells based on the Euclidean distance in the 50-dimensional principal components space, the main cell cluster was identified using the Louvain-Jaccard graph-based method. To classify all filtered cells, we set the clustering parameter resolution to 0.4 for the FindClusters function in Seurat. Next, the RunUMAP function in Seurat was used to reduce high dimensions into two dimensions (2D) for visualization. Finally, we ran the Seurat FindAllMarkers function with the default parameters to identify the genes specifically expressed in each cluster. The significance of the differences in gene expression was determined using the Wilcoxon rank sum test with Bonferroni correction, and cell types were manually annotated based on the cluster markers. To calculate the sample composition based on cell type, the number of cells for each cell type from each sample was counted. The counts were then divided by the total number of cells for each sample and scaled to 100% for each cell type.

**Cell-type subclustering**. The same functions described above were used to obtain the subclusters. After PCA to reduce dimensionality and build k-nearest neighbour graphs (k = 30) of the cells based on the Euclidean distance in the 50-dimensional principal components space, the main cell cluster was identified using the Louvain-Jaccard graph-based method. To classify all filtered cells, we set the clustering parameter resolution to 0.4 for the function FindClusters in Seurat. We also ran the Seurat FindAllMarkers function with the default parameters to identify the genes specifically expressed in each subcluster. To calculate the composition of different groups of cells for the subclusters, the numbers of cells in each group for each subcluster were counted.

**Tissue distribution of clusters**. We calculated the ratio of observed to expected cell numbers (Ro/e) for each cluster in different groups to quantify the group preference of each cluster[44]. The expected cell numbers for each combination of cell clusters and groups were obtained from the chi-square test.

**DEG analysis**. Differentially expressed gene (DEG) testing of two groups with each subcluster in MSC, pericyte and myeloid cells was performed using the FindMarkers function in Seurat. The significance of the differences in gene expression was determined using the Wilcoxon rank sum test with Bonferroni correction. The differences in genes between the two groups in each subcluster were determined based on the following criteria: (1) expressed in more than 10% of the cells within either or both groups; (2) $|\log_2\mathrm{FC}| > 0.25$; and (3) Wilcoxon rank sum test adjusted $P$ value < 0.05.

**GO enrichment analysis**. Enrichment scores ($p$ values) for selected numbers of GO annotations were calculated by the clusterProfiler (v3.14.3) R package with a hypergeometric statistical test with a threshold of 0.05, and the Benjamini–Hochberg method was used to estimate the false discovery rate (FDR). Enrichment was calculated for the input DEGs in the subcluster. The background was all the genes listed in the database of org.Hs.e.g.db. Finally, the barplot function was run for visualization.

**Inference of the cell differentiation state using trajectory analysis.** Trajectory analysis was performed using the Monocle2 R package (v2.14.0) to reveal cell state transitions in MSCs, pericytes and myeloid cells. The raw counts for cells in the intended cell types were extracted and normalized by the estimateSizeFactors and estimateDispersions functions with the default parameters. We used the differentially expressed genes (DEGs) of each subcluster in the intended cell types identified through the Seurat FindAllMarkers function with the default parameters to sort the cells in pseudotime order. Dimensional reduction and cell ordering were performed using the DDRTree method and the orderCells function. Finally, the plot cell trajectory function was run for visualization. The DEG changes along the pseudotime were also determined with the differentialGeneTest function with default parameters, and the DEGs with the adjusted $P$ value were visualized with the plot pseudotime heatmap function. Moreover, the DEG changes along the pseudotime in different branches were also determined with the BEAM function with the default parameter, and the DEGs with the adjusted $P$ value (qval <1e−4) were used to plot the heatmap showing the bifurcation expression patterns with the plot genes branched heatmap function.

**RNA velocity analysis**. Analysis of cellular trajectory by RNA velocity was performed using the package scVelo (v0.2.3) using dynamic modelling. To estimate the RNA velocities in the Myeloid and Pericyte subtypes, velocyto was used to distinguish unspliced and spliced messenger RNAs in each sample. The Python package scVelo was then used to recover the directed dynamic information by leveraging RNA-splicing information. Specifically, the data were first normalized using

the filter and normalize function with the following parameter settings: min shared counts 30, n top genes 2000. The first- and second-order moments were computed for velocity estimation using the moments function with the following parameter settings: n pcs 30, n neighbours 30. The velocity vectors were obtained using the velocity function of dynamical modelling. The velocities were projected into a lower-dimensional embedding using the velocity graph function, and k-nearest neighbour graphs of the cells were built using the neighbour function with the following parameter settings: n neighbours 10, n pcs 40. Finally, the velocities were visualized in the UMAP embedding using the umap function with default parameters.

**CytoTRACE analysis**. Myeloid and pericyte subtypes were extracted and uploaded to the CytoTRACE R package (v0.3.3). The raw counts for cells in the intended cell types were analysed by the CytoTRACE functions with the default parameters, and the output CytoTRACE score for each cell was then plotted on the UMAP.

**Regulatory network inference**. A single-cell regulatory network for each subcluster was constructed with the SCENIC python workflow. Specifically, GRNBoost2 (https://github.com/tmoerman/arboreto) in pySCENIC (v0.10.1) was applied to infer gene regulatory networks from raw count data. Then, potential direct-binding targets (regulons) were selected based on DNA-motif analysis. Finally, gene regulatory network activity for individual cells was identified. To find the different regulators for each subcluster in the groups, the regulon activity was averaged. A regulon-group heatmap was generated with the pheatmap package in R. In addition, the specific regulators of each subcluster were the union of predicted regulator lists and marker gene lists and were used to generate a specific regulator heatmap with the pheatmap package in R.

**Cell–cell ligand–receptor interaction analysis**. CellPhoneDB (v2.1.7) was applied for ligand–receptor analysis. The normalized counts and cell type annotation for each cell were imputed into CellPhoneDB to determine the potential ligand–receptor pairs. Interaction pairs with $p$ values > 0.05 were filtered out from further analysis. Three runs were performed on four groups of cells: all cells, all CF and MAT group cells, and all H-MAT group cells. The number of interactions between each pair of cell types was plotted by Cytoscape (v3.7.0). Meanwhile, selected specific pairs were plotted by the dot plot function in CellPhoneDB with default parameters.

**GWAS signature module score**. CD-associated locus information was downloaded from the NHGRI-EBI GWAS Catalogue (http://www.ebi.ac.uk/gwas), which reports the largest number of CD-associated genomic loci to date. The GWAS-related genes were extracted from 953 recorded loci and then searched for markers of each cell type. Finally, 31 GWAS genes were specifically related to cell types and used to calculate GWAS signature module scores for all cell types in different groups by the AddModuleScore function with default parameters in Seurat.

**Pathway signature module score**. To score individual cells for pathway activities, we used the AddModuleScore of Seurat. First, for each cell, we used an expression matrix to compute gene expression rankings in each cell with the AddModuleScore function, with default parameters. We chose pathways predominantly from the canonical pathways (CP) described in the Molecular Signatures Database (MSigDB) C2 collection (curated gene sets), and CP gene sets were then used to score each cell for each gene set and cell. Data visualization was performed using the ggplot2 R package.

**Gene set variation analysis (GSVA)**. Pathway analyses were predominantly performed on the HALLMARK described in the Molecular Signatures Database (MSigDB) H collection (curated gene sets),

exported using the GSEABase package (v1.48.0). To assign pathway activity estimates to individual cells, we applied GSVA using standard settings, as implemented in the GSVA package (v1.34.0). To assess differential activities of pathways (GSVA) between two subclusters of cells, we contrasted the activity scores for each cell using a generalized linear model in the limma package (v3.46.0). The results of these linear models were visualized using bar plots. For the latter, pathways that did not show significant changes (t value of GSVA score between −3 and 3) in any of the sets of cells contrasted in one analysis were marked grey.

**Single-cell metabolism quantification.** The scMetabolism package (v0.2.1) was used to compute the single-cell metabolic activity of MSCs and myeloid subtype cells. Specifically, the method was set as "VISION", and Kyoto Encyclopedia of Genes and Genomes (KEGG) metabolic gene sets were used for analysis[45]. The function Dot-Plot.metabolism was used for visualization. Metabolic pathway scores for each subcluster of cells were averaged and visualized using the ggplot2 R package.

### Statistics and reproducibility
Prism 8.0 software (GraphPad Software) was used to assess statistical significance. No statistical method was used to predetermine sample size. No data were excluded from the analyses. Cells with low-quality were excluded based on standard single-cell preprocessing procedures. The experiments were not randomized. The Investigators were not blinded to allocation during experiments and outcome assessment.

### Reporting summary
Further information on research design is available in the Nature Portfolio Reporting Summary linked to this article.

## Data availability
The datasets (raw data) generated in this study are available through the Sequence Read Archive (SRA), BioProject ID: PRJNA871957. The processed single cell count tables are provided in Gene Expression Omnibus; GSE215001. The Molecular Signatures Database (MSigDB) can be explored at https://www.gsea-msigdb.org/gsea/msigdb. Source data are provided with this paper.

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

## Acknowledgements

We thank the staff members of the Haijun Deng, Kai Sun and Tingyu Mou work teams from the Department of General Surgery of Nanfang Hospital of Southern Medical University (Guangzhou, China) for providing assistance in the specimen collection. We would like to thank Qiaobing Huang from Southern Medical University for their valuable input on the manuscript composition. We acknowledge the Ping Lan and Jia Ke work team from the Sixth Affiliated Hospital of Sun Yat-sen University (Guangzhou, China) for the provision of human tissue used in the study. We thank Lin Yin and Hui Zhang from YUANXIN Biotech Co., Ltd. (Guangzhou, China) for providing assistance with scRNA-seq bioinformatics analysis. We are grateful for the assistance provided by Xiqiang Hong from the Wuhan Metware Biotechnology Co., Ltd for their kind help with lipidomics analysis. This research was supported by the National Natural Science Foundation of China to L.B. and F.X. (81970451 and 82070534), Natural Science Foundation of Guangdong Province to L.B. and F.X. (2023A1515012537, 2021A1515010013 and 2021A1515012260), Clinical Research Startup Program of Southern Medical University by High-level University Construction Funding of Guangdong Provincial Department of Education to L.B. (LC2019ZD021), Clinical Research Program of Nanfang Hospital, Southern Medical University to L.B. (2020CR027 and 2018CR038) and Guangdong Science and Technology Project (2017B020209003). Fang Xie and Lan Bai contributed equally to this work.

## Author contributions

F.F.W. conceived the project, planned the experiments, generated the data on human mesenteric adipose tissue scRNA-seq and metabolomics, performed tissue processing and analyses, data analysis and biological interpretation, and prepared the manuscript; F.T.W. and Q.Z. conceived of and performed the in vitro experiments, supported materials, generated and analysed data, and contributed to manuscript preparation; X.L., J.Y.F. and D.Z. provided experimental assistance and performed computational analyses and interpretation; W.D.W. was involved in scRNA-seq investigation; Y.T., Y.B.L. and Q.Q.L. assisted with histological analysis; X.H.P. contributed to manuscript revision; K.S. procured human mesenteric adipose tissue; F.X. performed pathological assessments, provided funding and intellectual contributions and critically appraised the manuscript; L.B. supervised the study, contributed to the study conception, critically appraised the manuscript and provided funding. All authors have read and approved the manuscript.

## Competing interests

The authors declare no competing interests.
