## [Peer Review File · Nature Communications]

REVIEWER COMMENTS

Reviewer #1 (Remarks to the Author):

Creeping fat (CF) in Crohn's disease, a disease characterizing phenomenon, that is only incompletely understood, is the focus of this manuscript. By exploring CF by applying scRNA-seq as well as lipidomics adipogenic and pro-fibrotic properties have been examined. The authors show an up-regulation of the sphingolipid and glycerolipid metabolism. Furthermore, a novel CCL2+DPP4+ subset of MSC could be identified as a CF specific cell population and results in dystrophic adipocytes. Complementary in vitro experiments indicate that CCL20+CD14+ monocytes accumulate and IL-6 results in activation of DPP4+ MSC. Thus, the present work adds another puzzle piece to the literature contributing to the understanding of CF.

- 60 patients (Figure 1a) were observed to have CF, how was CF defined?
- How was CD-MAT distinct from CF (how was it identified surgically)?
- The results section is written in a blurry way, with regard to Figure 1, please specify clearly in the text from which samples (whole tissue, isolated adipocytes) the lipidomics were performed.
- The statement that CF has a low lipid metabolic activity is unclear, how is normal activity defined or what is characterizing healthy MAT? In line, why are the two suggested pathways providing a competitive advantage for smaller adipocytes?
- The cellular changes in CF as defined by sc analysis have been described and do. Not provide real novelty. The confirmative analysis by flow cytometry was this performed on CF samples, please specify in the results section and comment on the marker section (which is currently not mentioned at all in the main text).
- The identification of the key pathways is of interest (and in line with previous literature, in particular the ECM deposition has been a focus of recent work
- The novelty of this work is the definition of MSC subpopulations, however, the confirmation of preadipocyte state is not clear, normally this would demand for polarization of these cell populations towards adipocytes.
- The interaction of CCL2+ monocytes and MSC is intriguing and provides an additional function to this cell population that has numerous times been identified in CF. The role of IL-6 (previously described to be up-regulated in CF) is intriguing, how can the authors exclude that there is an additional factor playing a role?

- cMo-1s were more frequent in CF, the correlation with CDAI is often difficult (although present here), however, a correlation with inflammation on the tissue level would be more convincing (since CDAI is not a very precise marker for small intestinal disease).
- The distribution of MSC2 &3 seems to be specific (Figure 3) for CD but not necessarily for CF?
- The group of Ha et al. (Cell 2020) equally performed sc analysis of the CF, were you able to identify your subpopulations (MSC also in their data set)? This would be a strong validation.

Minor:

- The manuscript requires thorough proof-reading
- Abbreviations have only incompletely been introduced (or only on the methods section, but not when mentioned first in text), this should be addressed
- Abstract: What is implied by insufficient lipid metabolism of CF

Reviewer #2 (Remarks to the Author):

The authors characterized ex vivo adipocytes and mesenchymal stem cells from mesenteric fat in fibrostenotic CD, non-stenotic CD, and healthy controls.

They first observe that creeping fat is microscopically different from fat in other mesenteries (smaller and polypartitioned adipocytes, immune cell infiltration, ECM deposition). Then they show differences in several pro-adipogenic metabolic pathways compared to controls. They proceed to identify 3 subpopulations of mesenchymal stem cells possibly transitioning from the first to the third with a corresponding lower proliferative activity and higher adipogenic capacity.

Finally, they identify a subset of macrophages (cMo-1s) abundant in CF and suggest it's responsible for inducing MSC transition from subtype 1 to 3. Interestingly cMo-1s have a different receptor for IL-6 which could explain the complex role of IL-6 in the development of CD fibrosis.

Overall I found the work Wu and colleagues interesting and novel, and the methodology sound. From a clinician's point of view, clarification on sample number, selection, and patients' characteristics would be appreciated. It seems to me that the supplementary table with patients' characteristics refers only to a subset.

Muscularis propria cells are also believed to play a key role in fibrostenosis [Ren Mao et al. Gut, 2022]. Comment on how your model of CF development could fit previous observations on the extracellular matrix from activated muscle cells. Could cMo-1s activate both muscle cells and MSC, thus linking the two mechanisms?

Was there any correlation between the amount of creeping fat and the severity of the fibrosis or duration of inflammation?

Throughout the text, the numbers of samples/patients is not very clear to me. For example in the results section, line 93, "CF was observed in the great majority of strictured CD patients (59/60) and the most of fibrotic and stricturing intestine (118/133)." Is the second proportion referring to surgical specimens from the previous 60 patients? is it another cohort? Ileum vs colon? the only details I can find in the supplement refer to the 53 patients for lipid metabolomic analysis.

On the same point it's important to know if all specimens were from the ileum or there were also colons or anastomosis (meaning stenosis on previously resected tissue) and where the healthy controls came from. If available, any extra details on treatment status of patients would be useful.

How was the subset of patients for metabolic analysis chosen? It's important to clarify the criteria to rule out selection bias

Minor: There are several spelling and grammar slips throughout the text, please review carefully.

Reviewer #3 (Remarks to the Author):

In their manuscript entitled "A unique CCL2+DPP4+ 1 subset of mesenchymal stem cells expedites aberrant 2 formation of creeping fat in humans" Wu et al describe cellular and molecular changes occurring in Creeping fat (CF) of Crohn's disease (CD) patients by using electron microscopy, lipidomics as well as scRNA analyses of surgical resectates obtained from CD patients and controls. The authors thereby observe that adipocytes of Creeping Fat are hyperplastic and characterized by alterations in lipid storage and lipid metabolism. Furthermore, Wu et al perform an in-depth characterization of the stromal-vascular fraction of CF using scRNA sequencing indicating that CF features an enrichment of a specific subtype of differentiated mesenchymal stem cells (MSC) as well as an enrichment of IL-6

producing macrophages. In-vitro, the authors then show that the formation of differentiated MSC is dependent on Il-6 signaling, arguing in favor of an inflammation driven differentiation of mesenchymal stem cells ultimately changing the homeostasis of adipocytes and potentially driving CF towards a more fibrogenic phenotype.

The topic is definitely of high interest and the experiments are logic. However, the authors should address the following points:

1. The manuscript is sometimes very hard to follow and the authors should consider to rewrite several parts of the manuscript under supervision of a native speaker (especially the abstract is very difficult to read and confusing!)

2. The quality of figures should be improved and the authors should ensure that all graphs are correctly labels and that the labels can be read.

3. The cohort of patients is not well enough defined:

- the average age of the control group appears significantly older than the CD cohort and it is not clear how these differences in age might affect the differences observed in mesenchymal stem cells. It would be appreciated if the authors could compare the composition of fat residing mesenchymal cells between younger and older healthy controls by reanalyzing the existing RNA seq

- No BMI Values are provided for patients and controls. Is there any correlation between BMI and the presence of more terminally differentiated MSC? How does the BMI affect the intestinal inflammation score in CD patients and does the BMI affect the infiltration of CF with inflammatory lymphocytes and macrophages?

- No information on the anti-inflammatory therapies of CD patients is provided. It would be appreciated if the authors could specify what drugs the CD patient had received prior to surgery to exclude drug related effects on immune cell composition

4. The link between CF-specific MSC differentiation and development of hyperplastic adipocytes is currently missing. To this end, the authors should consider to perform snRNA sequencing of CF, which would allow the detection of both stromal vascular cells and adipocytes.

Response to reviewers

Reviewer comments

Reviewer #1:

Creeping fat (CF) in Crohn's disease, a disease characterizing phenomenon, that is only incompletely understood, is the focus of this manuscript. By exploring CF by applying scRNA-seq as well as lipidomics adipogenic and pro-fibrotic properties have been examined. The authors show an up-regulation of the sphingolipid and glycerolipid metabolism. Furthermore, a novel CCL2+DPP4+ subset of MSC could be identified as a CF specific cell population and results in dystrophic adipocytes. Complementary in vitro experiments indicate that CCL20+CD14+ monocytes accumulate and IL-6 results in activation of DPP4+ MSC. Thus, the present work adds another puzzle piece to the literature contributing to the understanding of CF.

Response:

Thank you for this positive summary of our work. We have carefully considered these points and revised the manuscript accordingly.

- 60 patients (**Figure 1a**) were observed to have CF, how was CF defined?

Response:

Thank you for your question regarding the definition of CF in our study involving 60 patients with Crohn's disease. CF is visually striking because inflammation in the intestine of CD patients is patchy rather than continuous[1]. As a result, CF is also observed in patches, extending like fingerlike projections gripping the inflamed

(involved) segments of the intestine[2]. The presence of CF was determined using visual confirmation during surgery and histopathological examination of surgical specimens[3]. In our study, we defined CF as the expansion of mesenteric adipose tissue (MAT) that wraps around the inflamed segments of the intestine in patients with Crohn's disease, especially the inflamed ileum with fibrosis and stricture, and covered up to 50% of the bowel circumference. Our criteria for CF identification were designed to be consistent with the literature and provided an accurate representation of the CF phenomenon in patients with CD[2]. All the samples in our study were taken from the hyperplastic MAT outside the inflamed and stenotic ileum and the adjacent uninvolved MAT. We have supplemented the definition in Methods section (see page 18, Methods: Patients and specimen preparation, lines 480-482).

1. Roda G, *et al.* Crohn's disease. *Nature reviews Disease primers* 6, 22 (2020).
2. Ha CWY, *et al.* Translocation of Viable Gut Microbiota to Mesenteric Adipose Drives Formation of Creeping Fat in Humans. *Cell* 183, 666-683.e617 (2020).
3. Atreya R, Siegmund B. Location is important: differentiation between ileal and colonic Crohn's disease. *Nature reviews Gastroenterology & hepatology* 18, 544-558 (2021).

- How was CD-MAT distinct from CF (how was it identified surgically)?

Response:

Thank you for your question regarding the distinction between CD-MAT and CF in our study. CD-MAT and CF are closely related phenomena observed in patients with Crohn's disease[1, 2]. While CF is characterized by the expansion of mesenteric adipose tissue wrapping around the inflamed and fibrotic segments of the intestine, CD-MAT refers to the mesentery adipose tissue that does not undergo morphological

alterations in response to the inflammatory process in Crohn's disease[1].

In our study, we differentiated between CD-MAT and CF using both imaging and surgical findings. CD-MAT was identified as mesenteric adipose tissue adjacent to the uninvolved intestine, which demonstrated distinct histological and molecular features compared to the affected mesenteric adipose tissue[1]. On the other hand, CF was visually observed as adipose tissue that had expanded and wrapped around the inflamed bowel segments[1]. Surgically, CD-MAT was distinguished from CF based on its location, appearance, and the presence of histopathological features (**Figure 1a** and **Supplementary Figure 1c**). We have supplemented the specific location of the specimens in the Methods section (see page 18, Methods: Patients and specimen preparation, lines 484-488) and updated the Supplementary table 1a. The location of the specimens in our research is almost consistent with the present study.

a

Involved segment with CF

Uninvolved segment

Figure 1. Characteristics, lipidomics and scRNA survey of human MAT in CD. a. Macroscopic characteristics of representative involved ileal segments with attached hyperplastic CF from CD patients.

Supplementary Figure 1. Differential lipidomics profiles of H-MAT and CF. c. H&E stained paired uninvolved and adjacent involved ileal segments (CD uMUC and iMUC, respectively) with attached uninvolved mesenteric adipose (CD MAT) and adjacent CF. Scale bars, 200 μ M.

1. Ha CWY, *et al.* Translocation of Viable Gut Microbiota to Mesenteric Adipose Drives Formation of Creeping Fat in Humans. *Cell* 183, 666-683.e617 (2020).
2. Schäffler A, Herfarth H. Creeping fat in Crohn's disease: travelling in a creeper lane of research? *Gut* 54, 742-744 (2005).

- The results section is written in a blurry way, with regard to Figure 1, please specify clearly in the text from which samples (whole tissue, isolated adipocytes) the lipidomics

were performed.

Response:

We thank the reviewers for the suggestions and apologize for any confusion caused by the unclear presentation of the results. As described in the Methods section, we performed lipidomics analysis on creeping fat (CF) and uninvolved mesenteric adipose tissue (CD-MAT) from patients with Crohn's disease and healthy mesenteric adipose tissue (H-MAT) from control subjects. Specifically, the whole-MAT samples were thawed on ice, followed by lipid extraction and liquid chromatography-tandem mass spectrometry (LC-MS/MS) analysis to quantify the lipid species. By analysing whole-tissue samples, we aimed to capture the complex interactions and lipid metabolites in CF. We have supplemented the details in the part of Results (see page 5, Results: Integrated quantification of the adipogenic milieu with histopathology and lipidomics for human hyperplastic MAT in CD, paragraph 2, line 110) and Methods section (see page 23, Methods: Quantitative lipid profiles, paragraph 1, lines 649-656)

- The statement that CF has a low lipid metabolic activity is unclear, how is normal activity defined or what is characterizing healthy MAT? In line, why are the two suggested pathways providing a competitive advantage for smaller adipocytes?

Response:

We thank the reviewer for raising this great point. We appreciate the opportunity to clarify our findings and provide more message and interpretation. We acknowledge that our statement regarding CF having low lipid metabolic activity may not have been

rigorous enough, and further experiments are needed to define metabolic activity[1]. In addition to the metabolite measurement mentioned in our paper, other experiments are necessary to fully illustrate metabolic activity[2-5], such as 1) energy metabolism measurement, which evaluates energy metabolism by measuring energy expenditure and production through indirect or direct methods; 2) biochemical parameter determination, including measurements of blood glucose levels, cholesterol levels, lactate levels, etc., which can reflect the biochemical state and metabolic health of tissues; 3) tissue section staining, a technique used to observe cell structures and subcellular structures related to metabolism, such as mitochondrial quantity and morphology; and 4) molecular biology experiments, such as gene expression studies and signalling pathway research, which can reveal the underlying mechanisms regarding tissue metabolism. In fact, the original intention of lipidomics in our research was to reveal the mechanism why CF presented as both pro-fibrotic and pro-adipogenic, specifically with the emergence of numerous nascent adipocytes. We defined "normal" or "healthy" activity as the metabolic profile observed in H-MAT samples obtained from individuals without inflammatory bowel disease. In our revised manuscript, we provide additional detailed descriptions of the differences in lipidomics profiles between healthy and CF adipose tissue, with the inclusion of H-MAT as a control (see page 5, Results: Integrated quantification of the adipogenic milieu with histopathology and lipidomics for human hyperplastic MAT in CD, paragraph 2, lines 114-121).

With respect to our contention that the two proposed pathways confer a competitive

advantage for smaller adipocytes, this is purely speculative based on both our lipidomic results and literature, and we recognize that this claim was imprecise. We have hence revised our statement accordingly (see page 5, Results: Integrated quantification of the adipogenic milieu with histopathology and lipidomics for human hyperplastic MAT in CD, paragraph 2, lines 121-124).

Supplementary Figure 1. Differential lipidomics profiles of H-MAT and CF.

d. Heatmaps of the differences between H-MAT and CF. Data in the heatmap are log₂ FC values for each individual lipid. The lipids are sorted within each subclass level by the increasing total number of carbons in the hydrocarbon chains and then the total

number of double bonds. e. DA score of the differences in KEGG pathway activities between H-MAT and CF. The size of the circle represents the number of differential metabolites in the pathway and the length of the line segment represents the absolute value of the DA score which indicates the degree of decrease or increase. f. The metabolite intensity enriched in glycerolipid metabolism and regulation of lipolysis in the adipocytes pathway of H-MAT, CD-MAT and CF. **p < 0.01, ***p < 0.001 and ****p < 0.0001. TG = triglycerides; HexCer =glucosyl- or galactosylceramides; DG = diglycerols; LPE = lysophosphatidylethanolamines; PE-P =Phosphatidylethanolamine; PC = phosphatidylcholines; Cer = ceramides; LPC = lysophosphatidylcholines; LPS= lysophosphatidylserines; LPE-P =lysophosphatidylethanolamines; LPI = lysophosphatidylinositols; PE = phosphatidylethanolamines.

1. Abulikemu A, *et al.* Silica nanoparticles aggravated the metabolic associated fatty liver disease through disturbed amino acid and lipid metabolisms-mediated oxidative stress. *Redox biology* 59, 102569 (2023).
2. Bartman CR, TeSlaa T, Rabinowitz JD. Quantitative flux analysis in mammals. *Nature metabolism* 3, 896-908 (2021).
3. Di Cesare F, *et al.* Lipid and metabolite correlation networks specific to clinical and biochemical covariate show differences associated with sexual dimorphism in a cohort of nonagenarians. *GeroScience* 44, 1109-1128 (2022).
4. Gam C, *et al.* Unchanged mitochondrial phenotype, but accumulation of lipids in the myometrium in obese pregnant women. *The Journal of physiology* 595, 7109-7122 (2017).
5. Bachem A, *et al.* Microbiota-Derived Short-Chain Fatty Acids Promote the Memory Potential of Antigen-Activated CD8(+) T Cells. *Immunity* 51, 285-297.e285 (2019).

- The cellular changes in CF as defined by sc analysis have been described and do. Not provide real novelty. The confirmative analysis by flow cytometry was this performed

on CF samples, please specify in the results section and comment on the marker section (which is currently not mentioned at all in the main text).

Response:

Thank you for your feedback on our paper. We appreciate your comment on the novelty of our sc analysis results and the need to clarify the confirmatory analysis by flow cytometry. We apologize for any lack of clarity in our initial description of the study methods.

To identify specific cell populations, we utilized a panel of markers for flow cytometry analysis, which comprised of CD45⁺ immune cells, CD31⁺ endothelial cells, CD31⁻CD45⁻CD146⁺ pericytes and CD31⁻CD45⁻CD146⁻ MSCs. Among these MSCs, we purified the CD26⁺CD142⁻CD54⁻ MSC1s, CD142⁺CD26⁻CD54⁻ MSC2s and CD54⁺CD26⁻CD142⁻ MSC3s. The specific markers used for identifying the CCL2⁺DPP4⁺ subset of MSCs were mentioned in the Methods section, and we have now included this information in the main text (see page 8, Results: MSC subpopulations are more vibrant and prone to adipogenesis in CF, paragraph 2, lines 198-204). Additionally, all flow cytometry gating strategies are shown in **Supplementary Figure 2b**.

We recognize that our description of the confirmatory analysis by flow cytometry was insufficiently clear, and we have addressed this issue in the revised manuscript. Specifically, we have added a more detailed description of the panel of markers used for

identification of those cell populations, including those corresponding to CF samples, and supplemented this information in the main text to provide a more comprehensive understanding of our methodology. Thank you again for your valuable suggestion.

Supplementary Figure 2. Dynamic restructuring of cells in CD.

b. CD45⁻ cells were stained with anti-CD31, anti-CD146, anti-DPP4, anti-ICAM1, and anti-CD142 antibodies. Endothelial cells (CD31⁺) were selected first, then pericytes (CD31⁻, CD146⁺) were selected, and MSC2s (CD146⁻, CD142⁺) were selected subsequently, followed by MSC1s (CD142⁻, ICAM1⁻, DPP4⁺) and MSC3s (CD142⁻, DPP4⁻, ICAM1⁺). FSC-A: forwards scatter area, FSC-H: forwards scatter height, SSC-H: side scatter height.

- The identification of the key pathways is of interest (and in line with previous literature, in particular the ECM deposition has been a focus of recent work

Response:

We appreciate the reviewer's recognition of the interest and relevance of the identified key pathways, particularly ECM deposition which has been a focus of recent work in

the field. Increasing evidence indicates that the ECM not only plays a structural role but also actively participates in disease initiation and progression[1,2]. Recent research has demonstrated that aberrant ECM remodelling is involved in mesenteric adipocyte dysfunction in CD, which is associated with the excessive infiltration of macrophages at least partly through TLR4-regulated ECM remodelling [3]. In addition, Ren Mao et al. reported that activated muscularis propria muscle cells secrete a distinct matrix, with increased amounts of the extracellular matrix component fibronectin which, through integrin $\alpha5\beta1$ - mediated signalling, induces the migration of preadipocytes out of mesenteric fat[4]. Besides, the ECM may play an active role in inflammation by modulating immune cell functions, including cell adhesion, in IBD[5]. Human intestinal myofibroblast-derived ECM in IBD binds a markedly enhanced number of T cells, which is dependent on collagen VI and integrin $\alpha v\beta1$ [5].

We agree that further investigation into the specific mechanisms involved in ECM deposition and the potential therapeutic implications are warranted.

1. Derkacz A, Olczyk P, Olczyk K, Komosinska-Vassev K. The Role of Extracellular Matrix Components in Inflammatory Bowel Diseases. *Journal of clinical medicine* 10, (2021).
2. Wu X, et al. Cellular and Molecular Mechanisms of Intestinal Fibrosis. *Gut and liver*, (2023).
3. Zuo L, et al. Aberrant Mesenteric Adipose Extracellular Matrix Remodelling is Involved in Adipocyte Dysfunction in Crohn's Disease: The Role of TLR-4-mediated Macrophages. *Journal of Crohn's & colitis* 16, 1762-1776 (2022).
4. Mao R, et al. Activated intestinal muscle cells promote preadipocyte migration: a novel mechanism for creeping fat formation in Crohn's disease. *Gut* 71, 55-67 (2022).
5. Lin SN, et al. Human intestinal myofibroblasts deposited collagen VI enhances adhesiveness for T cells - A novel mechanism for maintenance of intestinal inflammation. *Matrix biology : journal of the International Society for Matrix Biology* 113, 1-21 (2022).

- The novelty of this work is the definition of MSC subpopulations, however, the

confirmation of preadipocyte state is not clear, normally this would demand for polarization of these cell populations towards adipocytes.

Response:

Thank you for your comment regarding our work and the preadipocyte state. Our study aimed to characterize MSC subpopulations in patients with CD, and we also observed the formation of lipid droplets following adipogenic differentiation, which is a characteristic feature of preadipocytes transitioning into mature adipocytes [1,2]. While we did not provide direct evidence of fully matured adipocytes, the presence of lipid droplets suggests that the MSC subpopulations we identified are capable of differentiating into adipocytes under appropriate conditions, as validated by Schwalie *et al.* [3]. In this study, we performed magnetic sorting of DPP4⁺ MSCs (MSC1s), CD142⁺ MSCs (MSC2s) and ICAM1⁺ MSCs (MSC3s). Consistent with Merrick *et al.*[4], after induction of adipogenic differentiation, we quantified the lipid droplets by BODIPY staining and detected the expression of adipocyte-specific genes such as PPARG, FABP4, and CEBPA (**Figure 3h**), which are conducive to identifying the state of preadipocytes. We found that MSC3s had more lipid droplets and higher expression of those genes, suggesting that MSC3s were the "committed preadipocytes". Accordingly, we revised our manuscript to emphasize the observation of lipid droplets after adipogenic differentiation and to address concerns regarding confirmation of the preadipocyte state (see page 9, Results: MSC subpopulations are more vibrant and prone to adipogenesis in CF, paragraph 3, lines 219-220).

In future studies, we will consider the polarization of these cell populations towards adipocytes. The identification of these MSC subpopulations and their association with creeping fat (CF) in patients with Crohn's disease adds valuable information to the current understanding of the cellular and molecular mechanisms underlying CF development. Our findings contribute to the growing body of knowledge regarding MSCs and adipose tissue in the context of CD, which could serve as a foundation for future studies aimed at characterizing the functional roles of these subpopulations more precisely.

Figure 3. Characterization of the genomics and function of MSC subpopulations.

h. mRNA levels of osteocyte-specific, chondrocyte-specific and adipocyte-specific genes in MSC subpopulations exposed to osteogenic, chondrogenic and adipogenic differentiation inducers.

1. Rosen ED, Spiegelman BM. What we talk about when we talk about fat. *Cell* 156, 20-44 (2014).
2. Tang QQ, Lane MD. Adipogenesis: from stem cell to adipocyte. *Annual review of biochemistry* 81, 715-736 (2012).
3. Schwalie PC, *et al.* A stromal cell population that inhibits adipogenesis in mammalian fat depots. *Nature* 559, 103-108 (2018).
4. Merrick D, *et al.* Identification of a mesenchymal progenitor cell hierarchy in adipose tissue. *Science (New York, NY)* 364, (2019).

- The interaction of CCL2+ monocytes and MSC is intriguing and provides an additional function to this cell population that has numerous times been identified in CF. The role of IL-6 (previously described to be up-regulated in CF) is intriguing, how can the authors exclude that there is an additional factor playing a role?

Response:

We thank the reviewer for raising this interesting point. We share and appreciate your idea that there are multiple interactions between myeloid cells, especially monocytes/macrophages, and MSCs in various scenarios. Research has shown that oncostatin M secretion from macrophages enhances the osteogenic differentiation of MSCs during fracture healing[1]. In the environment of damaged lungs, lung macrophages secrete TNF- α and other cytokines to activate MSCs, promoting lung repair[2].

In our study, we identified IL-6 as a powerful factor for CF development, but it cannot be ruled out that other factors may play a role. We originally focused on IL-6 because it has been previously reported to be upregulated in CF and is known to have a role in adipogenesis and inflammation. Subsequently, gene profiling revealed that IL-6 was specifically overexpressed in MSC1-S1s and abnormally upregulated during the process of MSC1 differentiation into MSC2s and MSC3s in CF. Consistent with that, *ex vivo* validation suggests that IL-6 could activate MSC1s towards a pro-adipogenic phenotype.

In addition, we further functionally analyzed other candidate factors in MSC1 differentiation assays. CXCL2 or CXCL8 were respectively co-cultured with MSC1s, and the results showed that they were less effective than IL-6 (**Supplementary Figure 5**). Moreover, we tested a neutralizing antibody against IL-6 in the conditioned medium of cMo-1s, and found that the IL-6 neutralizing antibody greatly reduced the effect of activating MSC1s to a pro-adipogenic phenotype (**Figure 5I**).

Supplementary Figure 5. Phenotypic characterization of MSC1-S1. Flow cytometry showing ICAM1 (also known as CD54⁺) expression in MSC1 cells after stimulation with CXCL2 or CXCL8 (1 ng/mL or 5 ng/mL) for 12 hours.

Furthermore, recent elegant studies have revealed that IL-6 is considered a crucial target for IBD treatment[3, 4]. A recent phase 2 clinical trial published in JAMA demonstrated that olamkicept, a first-in-class, selective inhibitor of the sIL-6R/IL-6 complex, can increase the likelihood of clinical response in patients with IBD[5]. Another investigation published in Nature suggested that gp130 (also known as IL-6ST) blockade may benefit some CD patients with NOD2-driven fibrosis, potentially as a complement to anti-TNF therapy[6].

Figure 5. A distinct monocyte subset accumulates in CF and is pro-inflammatory and pro-adipogenic.

I. Flow cytometry showing ICAM1 (also known as CD54⁺) expression in MSC1s after co-cultured with cMo-1 culture medium with or without IL-6 neutralizing antibody for 12 hours.

In addition, as you indicated, future studies could investigate other potential factors involved in this interaction to gain a more complete understanding of the pathogenesis of CF.

1. Pajarinen J, *et al.* Mesenchymal stem cell-macrophage crosstalk and bone healing. *Biomaterials* 196, 80-89 (2019).
2. Jerkic M, Szaszi K, Laffey JG, Rotstein O, Zhang H. Key Role of Mesenchymal Stromal Cell Interaction with Macrophages in Promoting Repair of Lung Injury. *International journal of molecular sciences* 24, (2023).
3. Ito H. Treatment of Crohn's disease with anti-IL-6 receptor antibody. *Journal of gastroenterology* 40 Suppl 16, 32-34 (2005).
4. Danese S, *et al.* Randomised trial and open-label extension study of an anti-interleukin-6 antibody in Crohn's disease (ANDANTE I and II). *Gut* 68, 40-48 (2019).
5. Zhang S, *et al.* Effect of Induction Therapy With Olamkicept vs Placebo on Clinical Response in Patients With Active Ulcerative Colitis: A Randomized Clinical Trial. *Jama* 329, 725-734 (2023).
6. Nayar S, *et al.* A myeloid-stromal niche and gp130 rescue in NOD2-driven Crohn's disease. *Nature* 593, 275-281 (2021).

- cMo-1s were more frequent in CF, the correlation with CDAI is often difficult (although present here), however, a correlation with inflammation on the tissue level would be more convincing (since CDAI is not a very precise marker for small intestinal disease).

Response:

Thank you for your valuable suggestion. Your constructive suggestion was very helpful for us to improve the quality of this work. According to your opinion, we have supplemented the correlation analysis between cMo1s and inflammation at the tissue level, including the Simple Endoscopic Score for Crohn's Disease (SES-CD) and the expression of inflammatory genes, such as CXCL8 and IL-1 β (see page 12, Results: The pro-inflammatory monocyte subset accumulates in CF and promotes MSC1-S1 adipogenesis, paragraph 2, lines 312-313).

Supplementary Figure 7. Identifying unique myeloid subpopulations in human MAT.

j. Estimated proportions of monocytes in scRNA-seq data of CF from 6 individuals plotted against MCFI, SES-CD, as well as expression of IL1B and CXCL8 in strictured intestinal tissues. Linear regression and Spearman's correlation analysis with 95% CIs were conducted. $P < 0.05$ was considered significant.

- The distribution of MSC2 &3 seems to be specific (Figure 3) for CD but not necessarily for CF?

Response:

We thank the reviewer for raising this nice point. As the reviewer said, the distribution of three MSC subpopulations (MSC1-3) was similar between CF and CD-MAT. However, they are very different in terms of functionality. The results of GO analysis showed that the differences in pathway enrichment of MSC1-3 between CF and CD-MAT can be distinguished by regulation of the apoptotic signalling pathway, regulation of smooth muscle cell migration, regulation of epithelial cell proliferation, viral transcription, viral gene expression, respiratory electron transport chain, monophosphate metabolic process and platelet degranulation. Therefore, the three MSC subpopulations in CF are fundamentally different from those in CD-MAT (**Figure below**).

GO analysis of genes differentially enriched in MSC subpopulations between CD-MAT and CF.

- The group of Ha et al. (Cell 2020) equally performed sc analysis of the CF, were you able to identify your subpopulations (MSC also in their data set)? This would be a strong validation.

Response:

We thank the reviewer for the professional advice, which will significantly improve the quality of our article. Studies by Ha *et al.* provided profound insights into the

mechanism of CF [1]. We analyzed their datasets generated from the same platform (10X Genomics Chromium), including CF (GEO accession/patient ID: GSM4743752/A04 and GSM4743753/A16) and H-MAT (GSM4743754/D1 and GSM4743755/H1). We were able to identify our three subpopulations of MSCs in these scRNA-seq datasets generated by Ha *et al* (Figure below). This result provides additional validation for our findings and strengthens the idea that this MSC subset is specific to CF.

a. UMAP plots of CF and H-MAT were generated from Ha et al.'s scRNA-seq datasets, which distinguish individual cell clusters. The frequency of cell types is indicated by colored bars. **b.** Heatmap showing the expression of specific marker genes for MSC subpopulations.

1. Ha CWY, *et al.* Translocation of Viable Gut Microbiota to Mesenteric Adipose Drives Formation of Creeping Fat in Humans. *Cell* 183, 666-683.e617 (2020).

Minor:

- The manuscript requires thorough proof-reading

Response:

We thank the reviewer for the constructive advice. We have carefully revised our manuscript according to your opinion. We believe that the revised manuscript meets the requirements for publication.

- Abbreviations have only incompletely been introduced (or only on the methods section, but not when mentioned first in text), this should be addressed

Response:

We thank the reviewer for the suggestion. We apologize for any confusion caused by the incomplete introduction of abbreviations in our manuscript. We have revised the text to ensure that all abbreviations are properly introduced and defined at their first mention, as per the guidelines set forth by the journal. This will improve clarity and enhance the overall readability of the manuscript.

- Abstract: What is implied by insufficient lipid metabolism of CF

Response:

We thank the reviewer for raising this good point. We apologize for any ambiguity in our initial description of CF lipid metabolism, and we appreciate your attention to detail. We have since made revisions to clarify the matter in the revised manuscript (see page 2, Abstract, lines 39-41), and now describe the altered lipid metabolism in CF as primarily involving the regulation of lipolysis in adipocytes and glycerolipid

metabolism. We hope that these changes better address your concern, and we welcome any further feedback you may have.

Abnormal lipid metabolism in CF, as evidenced by the downregulation of metabolites involved in lipid metabolic pathways, has several potential implications in the context of Crohn's disease. First, it suggests that the adipose tissue in CF may have altered functional properties compared to healthy mesenteric adipose tissue (H-MAT)[1]. Altered lipid metabolism in CF may contribute to the dysregulated production of adipokines, which are known to play essential roles in modulating inflammation and immune responses[2]. This could exacerbate the local inflammatory environment in the affected intestinal segments, further aggravating the disease process[3]. Second, dysregulated lipid metabolism could potentially result in the formation of smaller, pro-inflammatory adipocytes that perpetuate the inflammatory process in Crohn's disease[4]. Last, changes in lipid metabolism could affect the interactions between adipose tissue and other cells in the local microenvironment, such as immune cells, endothelial cells and fibroblasts, potentially leading to alterations in tissue homeostasis and exacerbating disease progression[5,6]. In summary, insufficient lipid metabolism in CF may contribute to the altered functional properties of adipose tissue, exacerbate local inflammation, and affect cellular interactions within the tissue microenvironment, all of which may play a role in the pathogenesis of Crohn's disease.

1. Kredel LI, *et al.* T-cell Composition in Ileal and Colonic Creeping Fat - Separating Ileal from Colonic Crohn's Disease. *Journal of Crohn's & colitis* 13, 79-91 (2019).
2. Weidinger C, Ziegler JF, Letizia M, Schmidt F, Siegmund B. Adipokines and Their Role in Intestinal Inflammation. *Frontiers in immunology* 9, 1974 (2018).

3. Kredel LI, *et al.* Adipokines from local fat cells shape the macrophage compartment of the creeping fat in Crohn's disease. *Gut* 62, 852-862 (2013).
4. Yin Y, Xie Y, Ge W, Li Y. Creeping fat formation and interaction with intestinal disease in Crohn's disease. *United European gastroenterology journal* 10, 1077-1084 (2022).
5. Kredel LI, *et al.* T-cell Composition in Ileal and Colonic Creeping Fat - Separating Ileal from Colonic Crohn's Disease. *Journal of Crohn's & colitis* 13, 79-91 (2019).
6. Suau R, Pardina E, Domènech E, Lorén V, Manyé J. The Complex Relationship Between Microbiota, Immune Response and Creeping Fat in Crohn's Disease. *Journal of Crohn's & colitis* 16, 472-489 (2022).

Reviewer #2 :

The authors characterized *ex vivo* adipocytes and mesenchymal stem cells from mesenteric fat in fibrostenotic CD, non-stenotic CD, and healthy controls.

They first observe that creeping fat is microscopically different from fat in other mesenteries (smaller and polypartioned adipocytes, immune cell infiltration, ECM deposition). Then they show differences in several pro-adipogenic metabolic pathways compared to controls. They proceed to identify 3 subpopulations of mesenchymal stem cells possibly transitioning from the first to the third with a corresponding lower proliferative activity and higher adipogenic capacity.

Finally, they identify a subset of macrophages (cMo-1s) abundant in CF and suggest it's responsible for inducing MSC transition from subtype 1 to 3. Interestingly cMo-1s have a different receptor for IL-6 which could explain the complex role of IL-6 in the development of CD fibrosis.

Response:

We greatly appreciate your positive summary of our work. Your constructive and insightful comments have helped us greatly improve our manuscript. We have seriously considered your opinions and revised this manuscript accordingly.

Overall I found the work Wu and colleagues interesting and novel, and the methodology sound. From a clinician's point of view, clarification on sample number, selection, and patients' characteristics would be appreciated. It seems to me that the supplementary table with patients' characteristics refers only to a subset.

Response:

Thank you for your positive feedback on our work and for highlighting the need for clarification on sample number, selection, and patient characteristics. We understand that providing comprehensive information on the study participants is crucial for the interpretation of our findings from a clinical perspective.

In our study, we analyzed samples from a total of 60 patients with CD. The diagnosis of CD was established using typical criteria[1]. We also collected the information of 28 control subjects with non-inflammatory bowel disease who underwent ileal surgical resection. We have updated the new Table S1 presenting the sex, age, Montreal classification, preoperative parameters, preoperative medications (including antidiarrheals, 5-ASA, steroids, biologics, immunomodulators, and antibiotics) and so on of all patients mentioned in the manuscript.

We believe that these additions will provide a more comprehensive overview of the study population and help clinicians better understand the context of our findings.

1. Roda G, *et al.* Crohn's disease. *Nature reviews Disease primers* 6, 22 (2020).

Muscularis propria cells are also believed to play a key role in fibrostenosis [Ren Mao et al. *Gut*, 2022]. Comment on how your model of CF development could fit previous observations on the extracellular matrix from activated muscle cells. Could cMo-1s activate both muscle cells and MSC, thus linking the two mechanisms?

Response:

We thank the reviewer for raising this important point. This is an innovative hypothesis, and we believe that cMo-1s could potentially activate both muscle cells and MSCs, thus linking the two mechanisms. Ren Mao *et al.* confirmed that muscularis propria cells activated by TGF- β could promote the migration of preadipocytes out of mesenteric fat and differentiate into adipocytes, thus promoting the formation of CF[1]. Myeloid and stromal cells such as muscle cells and MSCs themselves, can function pathogenically, and in the same niche the intimate crosstalk between them can also play a pivotal role in the progression and treatment resistance of inflammatory diseases[2]. In addition, recent reports have noted that a dense network of resident macrophages populates the intestinal muscularis and that these monocytes/macrophages can be activated, subsequently initiating various inflammatory responses that lead to intestinal dysmotility[3,4,5]. As a result of intestinal microflora overgrowth, activated monocytes/macrophages may produce pro-inflammatory cytokines to induce a motility disorder in the dilated part of the ileum in a rat model of Hirschsprung's disease, and a similar phenomenon was found in rat model of TNBS-induced colitis and surgical induction of intestinal obstruction[6,7,8]. These studies show that monocytes/macrophages could closely interact with muscle cells. Therefore, we hypothesized that monocytes/macrophages may initiate various inflammatory responses that lead to activation of muscularis propria cells and ultimately promote the formation of CF.

Research has also shown that monocytes/macrophages can activate MSCs. During fracture healing, monocytes/macrophages secrete oncostatin M to enhance the differentiation of MSCs [9]. In damaged lungs, lung monocytes/macrophages can secrete TNF- α and other cytokines to activate MSCs to promote lung repair[10]. Our study focused on the role of MSCs in the formation of CF, but we acknowledge that other cell types and factors likely contribute to this process. In our scRNA-seq analysis, cMo-1s were mainly present in CF but absent in H-MAT and expressed inflammatory transcription factors and multiple genes. *In vitro*, the conditioned medium of cMo-1 cells efficiently activated MSC1s towards a pro-adipogenic phenotype.

Based on the above findings, we speculate that the inflammatory microenvironment in intestinal and mesenteric adipose tissue in CD patients can regulate mesenchymal cells such as muscle cells and MSCs. We believe that further studies investigating the interaction between cMo-1s, muscle cells, and other cell types involved in fibrostenosis and adipogenesis could provide valuable insights into the underlying mechanisms of CF development.

1. Mao R, *et al.* Activated intestinal muscle cells promote preadipocyte migration: a novel mechanism for creeping fat formation in Crohn's disease. *Gut* 71, 55-67 (2022).
2. Nayar S, *et al.* A myeloid-stromal niche and gp130 rescue in NOD2-driven Crohn's disease. *Nature* 593, 275-281 (2021).
3. Kalff JC, Schwarz NT, Walgenbach KJ, Schraut WH, Bauer AJ. Leukocytes of the intestinal muscularis: their phenotype and isolation. *Journal of leukocyte biology* 63, 683-691 (1998).
4. Mischopoulou M, D'Ambrosio M, Bigagli E, Luceri C, Farrugia G, Cipriani G. Role of Macrophages and Mast Cells as Key Players in the Maintenance of Gastrointestinal Smooth Muscle Homeostasis and Disease. *Cellular and molecular gastroenterology and hepatology* 13, 1849-1862 (2022).

5. Sui C, *et al.* Molecular and cellular mechanisms underlying postoperative paralytic ileus by various immune cell types. *Frontiers in pharmacology* 13, 929901 (2022).
6. Won KJ, *et al.* Increased smooth muscle contractility of intestine in the genetic null of the endothelin ETB receptor: a rat model for long segment Hirschsprung's disease. *Gut* 50, 355-360 (2002).
7. Kinoshita K, *et al.* Possible involvement of muscularis resident macrophages in impairment of interstitial cells of Cajal and myenteric nerve systems in rat models of TNBS-induced colitis. *Histochemistry and cell biology* 127, 41-53 (2007).
8. Yang NN, *et al.* Electroacupuncture ameliorates intestinal inflammation by activating α 7nAChR-mediated JAK2/STAT3 signaling pathway in postoperative ileus. *Theranostics* 11, 4078-4089 (2021).
9. Pajarinen J, *et al.* Mesenchymal stem cell-macrophage crosstalk and bone healing. *Biomaterials* 196, 80-89 (2019).
10. Jerkic M, Szaszi K, Laffey JG, Rotstein O, Zhang H. Key Role of Mesenchymal Stromal Cell Interaction with Macrophages in Promoting Repair of Lung Injury. *International journal of molecular sciences* 24 (2023).

Was there any correlation between the amount of creeping fat and the severity of the fibrosis or duration of inflammation?

Response:

We appreciate your interest in our work. We observed increased intestinal segment thickness and mesenteric adipose tissue expansion in fibrotic regions of patients with CD (**Supplementary Figure 1a**). To investigate the relationship between the amount of creeping fat and the severity of the fibrosis, we performed correlation analysis and observed a statistically significant correlation ($r = 0.8337$, $p = 0.0023$; **Supplementary Figure 1b**) between the maximal thickness of MAT and the maximal thickness of the small intestine in patients with CD. Similarly, a recent study developed a novel mesenteric creeping fat index (MCFI) using computed tomography (CT) in CD patients, which showed excellent correlation with the extent of fat wrapping and could accurately be used to differentiate the degree of intestinal fibrosis[1], indicating that increased an

amount of CF plays an important role in the fibrogenesis of CD.

In addition, our study focused on the dynamic changes in cellular components after MAT hypertrophy as well as the immediate inflammatory status. Unfortunately, we did not specifically design our study to further analyse the duration of inflammation. This was due to the fact that during treatment, some patients received antibiotics to varying degrees, which may have impacted this parameter. As a result, further quantitative data or analysis may be required to reveal any potential correlations. Furthermore, we conducted a literature review and found growing evidence suggesting that CF plays an active role in inflammation. CF is a rich source of TNF, interleukin-6, interleukin-10, and other pro-inflammatory cytokines, which could explain its link with intestinal inflammation in CD[2]. Recently, Rimola *et al.* researched magnetic resonance enterography predictors of long-term healing of severe inflammatory lesions in 58 CD patients starting anti-TNF therapies. The presence of CF was found to be a baseline-negative predictor of long-term healing of severe inflammation in the multivariable analysis[3]. These findings suggest a positive correlation between the severity of CF and the duration of inflammation.

In summary, CF is associated with the clinical activity of CD, and an increased amount of CF is likely to be positively associated with the severity of the fibrosis and duration of inflammation. We supplemented this information in the main text (see page 4, Results: Integrated quantification of the adipogenic milieu with histopathology and lipidomics

for human hyperplastic MAT in CD, paragraph 1, lines 99-105).

Supplementary Figure 1. The relationship between the amount of CF and the severity of fibrosis. a. The maximal thickness of the small intestine wall and MAT of CD patients (n=13). **b.** The maximal thickness of MAT or CF plotted against the maximal thickness of the corresponding intestinal wall from 11 individuals. Linear regression and Spearman’s correlation analysis with 95% CIs were conducted. $P < 0.05$ was considered significant. * $p < 0.05$, ** $p < 0.01$, *** $p < 0.001$ and **** $p < 0.0001$.

1. Li XH, *et al.* Degree of Creeping Fat Assessed by Computed Tomography Enterography is Associated with Intestinal Fibrotic Stricture in Patients with Crohn's Disease: A Potentially Novel Mesenteric Creeping Fat Index. *Journal of Crohn's & colitis* **15**, 1161-1173 (2021).
2. Mao R, *et al.* The Mesenteric Fat and Intestinal Muscle Interface: Creeping Fat Influencing Stricture Formation in Crohn's Disease. *Inflammatory bowel diseases* **25**, 421-426 (2019).
3. Rimola J, *et al.* Pre-treatment magnetic resonance enterography findings predict the response to TNF-alpha inhibitors in Crohn's disease. *Alimentary pharmacology & therapeutics* **52**, 1563-1573 (2020).

Throughout the text, the numbers of samples/patients is not very clear to me. For example in the results section, line 93, “CF was observed in the great majority of strictured CD patients (59/60) and the most of fibrotic and stricturing intestine

(118/133).” Is the second proportion referring to surgical specimens from the previous 60 patients? is it another cohort? Ileum vs colon? the only details I can find in the supplement refer to the 53 patients for lipid metabolomic analysis.

Response:

We thank the reviewers for this feedback and apologize for any confusion regarding the number of samples/patients. The presence of creeping fat was observed in 59 patients in the CD cohort, which consisted of 60 strictured CD patients. Since CD is a segmental lesion, there are multiple strictures in the affected intestinal segment of the same patient. The total number of strictured intestinal specimens collected from the same CD cohort that consisted of 60 strictured CD patients was 133, of which the presence of creeping fat was observed in 118 surgical specimens. We have revised the main text as follows: “CF was patchy rather than continuous, extending fingerlike projections gripping the diseased segments of intestine (**Figure 1a**), which we observed to be a consistent feature in the CD cohort that consists of 60 patients who underwent surgical resections due to strictures from CD (Supplementary table 1a). The presence of CF was observed in 59 patients in the CD cohort (59/60). Among the 60 patients with CD, 133 strictured intestinal segments were collected, of which the presence of CF was observed in 118.” (see page 4, Results: Integrated quantification of the adipogenic milieu with histopathology and lipidomics for human hyperplastic MAT in CD, paragraph 1, lines 95-100).

CF wraps specifically around sites of intestinal inflammation, primarily in the small bowel and most often the ileum[1]. In addition, CF may also expand in the colon when the ileocecal region is involved[2]. The samples we used for scRNA-seq were all taken from the ileum attached MAT. Regarding the location of the specimens, we analyzed both ileal and colonic specimens in the study, and the details of the specimen location are included in the new **Supplementary Table 1a**.

1. Ha CWY, *et al.* Translocation of Viable Gut Microbiota to Mesenteric Adipose Drives Formation of Creeping Fat in Humans. *Cell* **183**, 666-683.e617 (2020).
2. Atreya R, Siegmund B. Location is important: differentiation between ileal and colonic Crohn's disease. *Nature reviews Gastroenterology & hepatology* **18**, 544-558 (2021).

On the same point it's important to know if all specimens were from the ileum or there were also colons or anastomosis (meaning stenosis on previously resected tissue) and where the healthy controls came from. If available, any extra details on treatment status of patients would be useful.

Response:

We thank the reviewer for the suggestion. We apologize for any confusion regarding the specimen location and patient characteristics in our manuscript. To address this concern, we have updated **Supplementary Table 1a** to include the relevant demographic and clinical information for all the patients involved in the study. This includes details such as age, sex, BMI, disease duration, disease location, disease behaviour, and medication

use. Forty-eight specimens from CD patients were derived from the ileum, 8 were from anastomotic lesions, and four were from the colon. CF may also expand in the colon when the ileocecal region is involved. Sixteen specimens from non-IBD patients were from the ileum, and 11 specimens were from the colon. In addition, scRNA-seq was performed on six patients with ileum-CD and two control subjects with ileum affected. The location of the lesion is listed in Supplementary Table 1a to provide greater transparency. We have revised the methods (see page 18, Methods: Patients and specimen preparation, paragraph 1, lines 484-488). Besides, we also provided detailed information on the treatment status of patients in Supplementary Table 1a. In our study, CD patients received a range of medications, including antidiarrheals, 5-ASA, steroids, biologics, immunomodulators, and antibiotics. We listed the medications used within three months before surgery.

Once again, we apologize for any confusion and hope this additional information clarifies our methods and patient characteristics.

How was the subset of patients for metabolic analysis chosen? It's important to clarify the criteria to rule out selection bias.

Response:

We thank the reviewer for raising this good point. We apologize for any confusion regarding the selection of patients for lipidomics. Initially, we evaluated the changes in CF only by surgical and endoscopic appearance and histological examination. To further explore the cause of increased adipocyte formation in CF, we conducted lipidomics on

both CF and CD-MAT samples from CD patients, as well as H-MAT samples from control subjects. The patients for metabolic analysis were consecutively selected, and all available samples were collected in chronological order. Only samples that met our quality control criteria were included for the lipidomics experiment. These criteria included having sufficient sample mass for analysis and ensuring that the tissue was not significantly degraded or contaminated. There were no specific selection criteria based on patient characteristics or clinical status. Our lipidomic analysis revealed differences in lipid metabolism in CF compared to H-MAT. Subsequently, to clarify these differences at the cellular level, we performed scRNA-seq analysis of CF, CD-MAT and H-MAT. We understand the importance of transparency in study design and selection criteria, and have supplemented this information more clearly in the Methods section (see page 19, Methods: Patients and specimen preparation, paragraph 1, lines 484-488) and the new Supplementary Table 1b.

Minor: There are several spelling and grammar slips throughout the text, please review carefully.

Response:

Thank you for pointing this out. We have revised the manuscript and polished it using American Journal Experts recommended by your journal.

Reviewer #3:

In their manuscript entitled "A unique CCL2+DPP4+ 1 subset of mesenchymal stem cells expedites aberrant formation of creeping fat in humans" Wu et al describe cellular and molecular changes occurring in Creeping fat (CF) of Crohn's disease (CD) patients by using electron microscopy, lipidomics as well as scRNA analyses of surgical resectates obtained from CD patients and controls. The authors thereby observe that adipocytes of Creeping Fat are hyperplastic and characterized by alterations in lipid storage and lipid metabolism. Furthermore, Wu et al perform an in-depth characterization of the stromal-vascular fraction of CF using scRNA sequencing indicating that CF features an enrichment of a specific subtype of differentiated mesenchymal stem cells (MSC) as well as an enrichment of IL-6 producing macrophages. In-vitro, the authors then show that the formation of differentiated MSC is dependent on Il-6 signaling, arguing in favor of an inflammation driven differentiation of mesenchymal stem cells ultimately changing the homeostasis of adipocytes and potentially driving CF towards a more fibrogenic phenotype.

Response:

We are grateful to the reviewer for this positive summary of our work. We have carefully considered the main points of this study again and revised some manuscript content accordingly. We would appreciate if you could review it again.

The topic is definitely of high interest and the experiments are logic. However, the authors should address the following points:

1. The manuscript is sometimes very hard to follow and the authors should consider to rewrite several parts of the manuscript under supervision of a native speaker (especially the abstract is very difficult to read and confusing!)

Response:

We thank the reviewer for pointing this out. We appreciate your comments. We have taken your suggestions on board and rewrote certain parts of the manuscript with the assistance of a native English speaker to improve the overall clarity and readability. Our manuscript was polished using American Journal Experts as recommended by *Nature Communications*. We also took a close look at the abstract and rewrote it to ensure that it accurately conveys our research findings in a clear and concise manner.

2. The quality of figures should be improved and the authors should ensure that all graphs are correctly labels and that the labels can be read.

Response:

We thank the reviewer for the valuable feedback. We acknowledge that the quality of the figures could be improved, and we revised the graphics to ensure that all graphs were correctly labelled and that the labels were readable. We appreciate your attention to detail and your suggestions for improving the quality of our manuscript.

3. The cohort of patients is not well enough defined:

- the average age of the control group appears significantly older than the CD cohort and it is not clear how these differences in age might affect the differences observed in

mesenchymal stem cells. It would be appreciated if the authors could compare the composition of fat residing mesenchymal cells between younger and older healthy controls by reanalyzing the existing RNA seq

Response:

We thank the reviewer for the suggestions. We apologize for any confusion regarding the age differences between the control group and CD cohort. Although the average age of the control group seems to be older than that of the CD group in our study, Welch statistics showed no significant difference between the two groups ($p=0.3459$). We investigated the possibility of comparing the composition of fat-residing mesenchymal stem cells between younger and older healthy controls by reanalyzing the existing RNA-seq data (GSE176171)[1]. We selected seven datasets (including GSM5359325, GSM5359330, GSM5820679, GSM5820680, GSM5820681, GSM5820682 and GSM5820683) that matched our research for analysis, and the results showed that age differences had no effect on the composition of fat-residing mesenchymal stem cells (Figure below).

Estimated proportions of MSC1, MSC2 and MSC3 in CF samples from 6 CD patients plotted against age. Linear regression and Spearman's correlation analysis with 95% CIs

were conducted. $P < 0.05$ was considered significant.

1. Emont MP, *et al.* A single-cell atlas of human and mouse white adipose tissue. *Nature* 603, 926-933 (2022).

- No BMI Values are provided for patients and controls. Is there any correlation between BMI and the presence of more terminally differentiated MSC? How does the BMI affect the intestinal inflammation score in CD patients and does the BMI affect the infiltration of CF with inflammatory lymphocytes and macrophages?

Response:

We thank the reviewer for raising these important points. The BMI values of all the patients and controls in the study are listed in the new Supplementary table 1a. To test the relationship between BMI and the presence of more terminally differentiated MSCs, we performed correlation analysis. The results indicated that no significant correlation existed between these two variables in scRNA-seq and flow cytometry data (Spearman's correlation, $r=0.2571$, $p = 0.6583$ and $r=0.6768$, $p = 0.1556$, **Figure below a and b**).

Regarding the impact of BMI on the intestinal inflammation score, Singh et al. reported that although obesity, particularly visceral adiposity, seems to promote intestinal inflammation from a pathophysiological perspective, epidemiological studies implicate that obesity in the development of IBD are limited[1]. Furthermore, a high BMI was not consistently associated with an increased prevalence of CD-related adverse outcomes[2].

The effect of BMI on the intestinal inflammation score in CD patients remains to be studied.

In addition, with our data, a statistical analysis showed no significant correlation between BMI and the infiltration of CF with inflammatory lymphocytes (Spearman's correlation: $r=0.2000$ $p = 0.7139$, **Figure below c**) and macrophages (Spearman's correlation: $r=0.6000$, $p = 0.2417$, **Figure below d**).

Taking together, in this study, we focused on characterizing the cellular and molecular changes in CF of CD patients and identifying the subset of MSCs contributing to the aberrant formation of CF. However, we acknowledge that further studies may be needed to explore the impact of BMI on these factors in the context of CD, and we will explore this issue in more detail in the future.

Correlation analysis between cell proportions and BMI. **a.** Estimated proportions of MSC3 in CF samples from 6 CD patients by scRNA-seq plotted against BMI. **b.** Estimated proportions of MSC3 in CF samples from 6 CD patients by flow cytometry plotted against BMI. **c.** Estimated proportions of lymphocytes in stromal vascular cells (SVCs) from CF samples of 6 patients with CD plotted against BMI. **d.** Estimated proportions of macrophages in SVC from CF samples of 6 patients with CD plotted against BMI. Linear regression and Spearman's correlation analysis with 95% CIs were conducted. $p < 0.05$ was considered significant.

1. Singh S, Dulai PS, Zarrinpar A, Ramamoorthy S, Sandborn WJ. Obesity in IBD: epidemiology, pathogenesis, disease course and treatment outcomes. *Nature reviews Gastroenterology & hepatology* 14, 110-121 (2017).

2. Seminerio JL, *et al.* Impact of Obesity on the Management and Clinical Course of Patients with Inflammatory Bowel Disease. *Inflammatory bowel diseases* 21, 2857-2863 (2015).

- No information on the anti-inflammatory therapies of CD patients is provided. It would be appreciated if the authors could specify what drugs the CD patient had received prior to surgery to exclude drug related effects on immune cell composition

Response:

We thank the reviewers for this comment, and we agree that information on the anti-inflammatory therapies received by CD patients before surgery would be valuable to include in the manuscript. We have provided the anti-inflammatory therapies for all the patients with CD involved in the study in the new Table S1. The added details include including antidiarrheals, 5-ASA, steroids, biologics, immunomodulators, and antibiotics usage within 3 months. We have added this information to the Methods section of the manuscript to provide greater transparency (see page 19, Methods: Patients and specimen preparation, paragraph 1, lines 484-488) Although we did not observe significant differences in immune cell composition between CD patients with or without treatment in our scRNA-seq analysis, we cannot fully exclude the possibility of drug-related effects on immune cell composition.

Effect of drugs on the immune cell composition in CD patients

Cell type	Untreated CD patient's cell ratio(% ,mean±SD) (n=3)	Treated CD patient's cell ratio(% ,mean±SD) (n=3)	p value
Macrophages	1.90 ± 0.80	1.31 ± 1.79	0.627
Monocytes	4.28 ± 4.40	3.45 ± 2.28	0.787
Neutrophil cells	0.65 ± 0.35	3.56 ± 5.70	0.470

B cells	2.15 ± 1.51	19.46 ± 16.74	0.149
pDCs	0.12 ± 0.10	0.44 ± 0.31	0.166
cDCs	0.61 ± 0.27	0.61 ± 0.60	0.987
Plasma cells	1.10 ± 0.62	1.52 ± 1.71	0.709
Mast cells	0.41 ± 0.30	0.85 ± 0.69	0.375
NK cells	4.67 ± 6.31	2.01 ± 1.28	0.544
T cell 1	15.56 ± 8.76	18.78 ± 14.49	0.758
T cell 2	10.76 ± 4.18	25.92 ± 12.39	0.115

Data analysis was performed using Fisher's precision probability test.

4. The link between CF-specific MSC differentiation and development of hyperplastic adipocytes is currently missing. To this end, the authors should consider to perform snRNA sequencing of CF, which would allow the detection of both stromal vascular cells and adipocytes.

Response:

Thank you for your suggestion regarding the use of single-nucleus RNA sequencing (snRNA-seq) to better understand the link between CF-specific MSC differentiation and the development of hyperplastic adipocytes. We acknowledge that snRNA-seq could provide valuable insights into the cellular dynamics within CF, including both stromal vascular cells and adipocytes. Although we attempted snRNA-seq in our study, we encountered technical difficulties, and failed in obtaining an reliable data. For this particular tissue, snRNA-seq seems indeed somehow a challenge. Indeed, a comprehensive search of public databases did not yield any relevant human/mouse mesenteric adipose tissue data for snRNA-seq. On the other hand, as there is currently no recognized animal model for CF[1], we are unable to conduct *in vivo* validation experiments to track the fate and differentiation of MSC1-S1s by using cell lineage

tracing. Nonetheless, we were able to perform *in vitro* experiments on primary MSCs, which allowed us to investigate their adipogenic differentiation potential. Specifically, we observed the speed and morphology of lipid droplet formation and assessed the expression of adipocyte markers (**Figure 3**).

We appreciate your valuable input and, moving forwards to next project, we will explore these alternative methods to better understand the relationship between CF-specific MSC differentiation and the development of hyperplastic adipocytes.

1. Ha CWY, *et al.* Translocation of Viable Gut Microbiota to Mesenteric Adipose Drives Formation of Creeping Fat in Humans. *Cell* 183, 666-683.e617 (2020).

REVIEWERS' COMMENTS

Reviewer #1 (Remarks to the Author):

no further points

Reviewer #2 (Remarks to the Author):

I'm satisfied with the changes and clarifications the authors provided. They fully addressed mines and others' comments and improved the manuscript. No additional issues from my side.

Reviewer #3 (Remarks to the Author):

The manuscript has significantly improved and the reviewer has therefore no further major points that would need to be addressed.